# The Hsp70 homolog Ssb affects ribosome biogenesis via the TORC1-Sch9 signaling pathway

Kaivalya Mudholkar[1], Edith Fitzke[1], Claudia Prinz[1], Matthias P. Mayer [2] & Sabine Rospert [1,3]

The Hsp70 Ssb serves a dual role in de novo protein folding and ribosome biogenesis; however, the mechanism by which Ssb affects ribosome production is unclear. Here we establish that Ssb is causally linked to the regulation of ribosome biogenesis via the TORC1-Sch9 signaling pathway. Ssb is bound to Sch9 posttranslationally and required for the TORC1-dependent phosphorylation of Sch9 at T737. Also, Sch9 lacking phosphorylation at T737 displays significantly reduced kinase activity with respect to targets involved in the regulation of ribosome biogenesis. The absence of either Ssb or Sch9 causes enhanced ribosome aggregation. Particularly with respect to proper assembly of the small ribosomal subunit, *SSB* and *SCH9* display strong positive genetic interaction. In combination, the data indicate that Ssb promotes ribosome biogenesis not only via cotranslational protein folding, but also posttranslationally via interaction with natively folded Sch9, facilitating access of the upstream kinase TORC1 to Sch9-T737.

[1] Institute of Biochemistry and Molecular Biology, ZBMZ, Faculty of Medicine, University of Freiburg, D-79104 Freiburg, Germany. [2] Center for Molecular Biology of Heidelberg University (ZMBH), DKFZ-ZMBH-Alliance, D-69120 Heidelberg, Germany. [3] BIOSS Centre for Biological Signaling Studies, University of Freiburg, D-79104 Freiburg, Germany. Correspondence and requests for materials should be addressed to S.R. (email: sabine.rospert@biochemie.uni-freiburg.de)

C haperones of the Hsp70 family consist of a highly conserved 45 kDa N-terminal ATPase domain, a 15 kDa peptide binding domain, and a 10 kDa variable C-terminal domain. Via the peptide binding domain, Hsp70s dynamically interact with extended segments of substrate polypeptides. The cycle of substrate binding and release is controlled via the ATPase domain, which alternates between a low-affinity ATP, and a high-affinity ADP-bound state[1]. Yeast cells possess a specialized Hsp70 homolog termed Ssb, which is encoded by two nearly identical genes termed *SSB1* and *SSB2*[2, 3]. Ssb is distinguished from other Hsp70 homologs by its C-terminal domain, which allows it to directly interact with the ribosome in close proximity to the ribosomal tunnel exit[4, 5]. Together with its heterodimeric J-domain cochaperone, the ribosome-associated complex (RAC) consisting of Zuo1 and Ssz1, Ssb facilitates the folding of newly synthesized polypeptides emerging from the ribosomal tunnel[2, 6, 7]. However, it was recently discovered that the ability of Ssb to interact with ribosomes is not required to promote wild-type growth[4, 5, 7] and that the same applies to RAC[8]. Recent evidence suggests that the cytosolic pool of Ssb, in concert with RAC, which is the only J-domain partner of Ssb, interacts with a small set of proteins posttranslationally, not to assist de novo folding, but rather to facilitate proper regulation of posttranslational modifications[7, 9–11].

Yeast strains lacking Ssb or RAC contain reduced levels of assembled ribosomal particles[2, 3, 9, 12]. Moreover, in the absence of a functional Ssb/RAC system specifically ribosomal proteins and ribosome biogenesis factors display an enhanced tendency to aggregate[12, 13], accumulate in the nucleus[12], and cells suffer from severe defects in rRNA processing[12, 14]. These observations suggest that Ssb is a component of the ribosome assembly machinery[6, 12]. However, to date Ssb did not appear in screens, instrumental for the identification of more than 200 factors involved in ribosome biogenesis[15, 16] and it appears that more dedicated chaperones prevent the aggregation of ribosomal proteins[17].

When yeast cells grow logarithmically on glucose, ribosome biogenesis occurs at maximal speed due to the positive control of the target of rapamycin complex 1 (TORC1). Inhibition of TORC1 with rapamycin blocks the transcription of Pol I-dependent rRNA genes, Pol II-dependent ribosomal protein genes (RP genes), Pol III-dependent rRNA, and also the processing of 35 S rRNA[18–21]. The AGC kinase Sch9 is the central TORC1-effector regulating ribosome biogenesis[18, 19, 21–24]. In glucose-rich conditions Sch9 is activated via PDK-dependent phosphorylation of T570 as well as TORC1-dependent phosphorylation of five residues within the Sch9 C terminus (Sch9-CT)[18, 19, 24] (Fig. 1a). Glucose depletion leads to a drop of TORC1-dependent phosphorylation at the Sch9-CT[24, 25] and provokes massive changes in the transcriptome, which lead to a significant reduction of ribosomal particles[18–21]. One well characterized target of Sch9, which directly affects ribosome biogenesis is the transcriptional repressor Maf1[18, 26, 27]. In glucose-rich conditions Sch9 keeps Maf1 in a highly phosphorylated state, which is mainly localized in the cytosol. When Maf1 becomes dephosphorylated it enters the nucleus and inhibits Pol III activity[23, 26, 28].

Here we show that the Hsp70 homolog Ssb interacts with the kinase Sch9 in an ATP-sensitive manner and is required for the TORC1-dependent phosphorylation of one specific residue within the Sch9-CT. As a consequence, in cells lacking Ssb, Sch9 is inactive with respect to its direct target Maf1 and also with respect to the heat shock transcription factor Hsf1, which our data establish as a Sch9 target. Ribosome aggregation is not confined to cells lacking Ssb, but also occurs in cells lacking Sch9, indicating that Sch9 activity is required to prevent ribosome aggregation. Moreover, a permanently active Sch9 mutant partially suppresses ribosome aggregation in cells lacking Ssb. These findings fundamentally change the concept of Ssb's function in ribosome biogenesis. With respect to ribosome biogenesis, Ssb does not only act via its general foldase activity towards newly synthesized polypeptides, but also ensures proper signaling via the TORC1-Sch9 pathway in a posttranslational manner.

## Results

**In-depth dissection of Sch9 phosphorylation.** To understand the complex phosphorylation pattern of Sch9 (Fig. 1a), we analyzed differently phosphorylated species of the kinase via the phos-tag gel electrophoresis system[29]. In extracts derived from glucose-grown wild-type cells three major species of Sch9 were resolved via phos-tag gels. Sch9-0 (S0) represented the non-phosphorylated species co-migrating with phosphatase treated Sch9 (Fig. 1b). Sch9-1 (S1) represented a Sch9 species phosphorylated at T570, because i) S1 was recognized by an antibody specific for Sch9 phosphorylated at T570 (α-T570-Pi)[24] (Fig. 1c), and ii) S1 was absent in a Sch9-T570A mutant (Fig. 1d). Sch9-3 (S3) migrated as a poorly resolved double band (Fig. 1b–e), of which the upper band was i) detected by α-T570-Pi (Fig. 1c) and ii) was absent in the Sch9-T570A mutant (Fig. 1d).

Cycloheximide treatment shifted the bulk of S0/S1 to the S3 doublet, while treatment with rapamycin induced a collapse of Sch9 to S0/S1 (Fig. 1b, +CHX and +rapa). Rapamycin inhibits TORC1, while sublethal doses of cycloheximide activate TORC1[19, 24], indicating that the S3 doublet represented Sch9 species phosphorylated in a TORC1-dependent manner. Moreover, in the Sch9-5A mutant, in which the TORC1-dependent sites at the C terminus (Fig. 1a) were replaced with alanines[24], S3 was absent and only S0 and S1 were detected (Fig. 1c and Supplementary Fig. 1a). Sch9-5A S1 was recognized by α-T570-Pi (Fig. 1c) confirming that T570 phosphorylation was independent of TORC1-dependent phosphorylation at the Sch9-CT[24]. A Sch9 mutant carrying the T570A as well as the 5A-mutations (termed Sch9-T570A/5A) co-migrated with the S0 species (Fig. 1e, f). The combined data indicate that S0 was non-phosphorylated, S1 was phosphorylated at T570 within the activation loop, S3 was phosphorylated at TORC1-dependent sites within the Sch9-CT, and the upper band of the S3 doublet was in addition phosphorylated at T570 (Fig. 1f, left part).

Next we analyzed the phosphorylation pattern of Sch9 after brief (10 min) glucose depletion. Consistent with a drop of TORC1-dependent phosphorylation[24, 25], the Sch9 S3 doublet disappeared and a prominent, faster migrating doublet termed Sch9-2 (S2) was formed, which was phosphorylated at the PDK-dependent residue T570 (Fig. 1c). A comparison of the Sch9 and Sch9-5A phosphorylation pattern after glucose depletion revealed that S2 migrated more slowly compared to Sch9-5A-S2* (S2*) in phos-tag gels, indicating that at least one of the TORC1-dependent residues within Sch9 had retained phosphorylation (Supplementary Fig. 1a). In addition to the S2 species a hyper-phosphorylated species termed Sch9-4 (S4) was formed upon glucose depletion (Fig. 1c). Analysis employing α-T570-Pi and the Sch9-5A mutant revealed that S4 was phosphorylated at the PDK-dependent as well as TORC1-dependent residues (Fig. 1c and Supplementary Fig. 1a). Thus, upon glucose depletion, most, but not all, of the TORC1-dependent phosphorylations were removed from Sch9, however, at the same time Sch9 received phosphorylations at one, or more, residue(s) distinct from the PDK and TORC1 target sites. To confirm this we employed the Sch9-T570A/5 A mutant, which was non-phosphorylated in glucose-rich conditions (Fig. 1e).

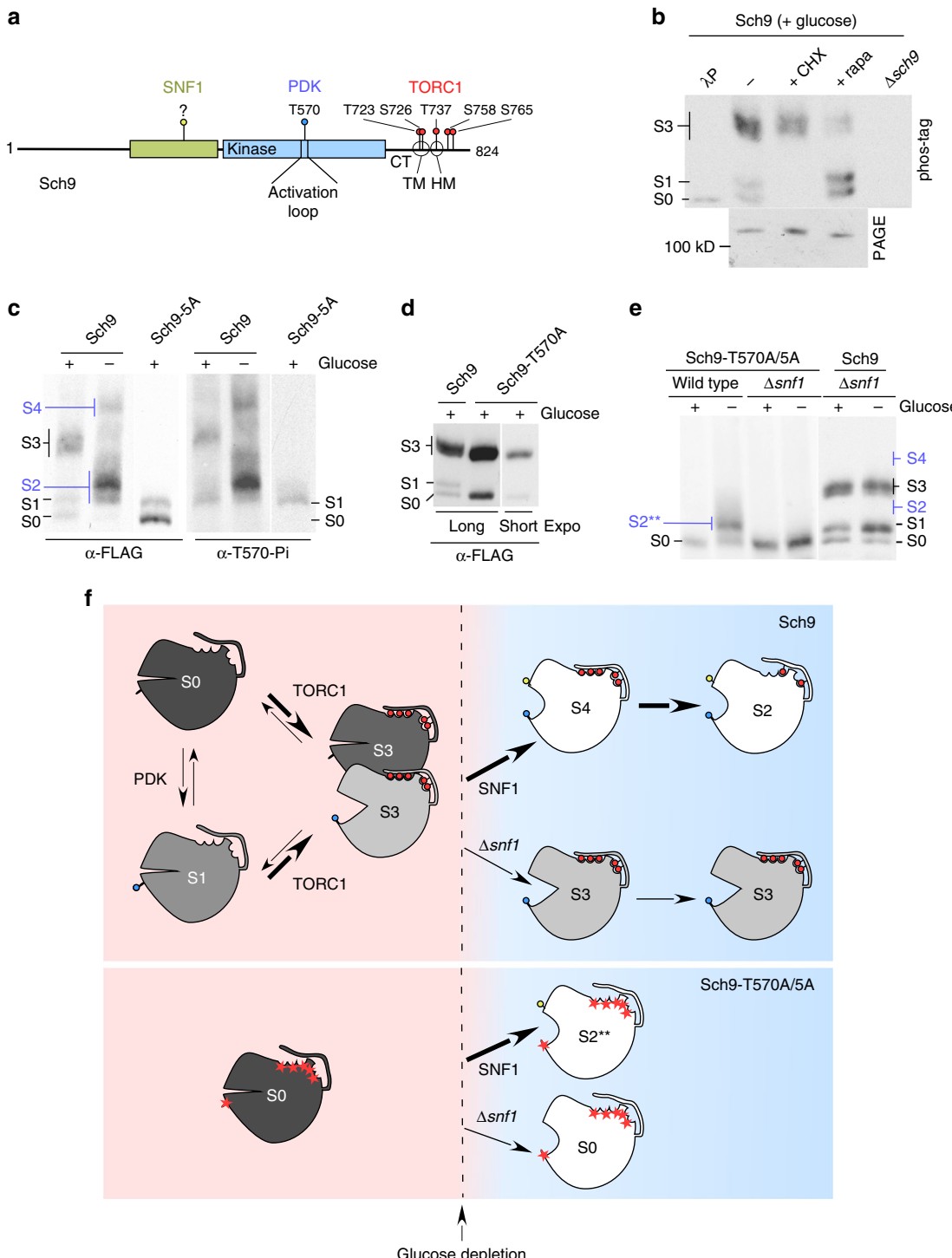

**Fig. 1** Phosphorylation pattern of full length Sch9 in glucose-rich and glucose-starved conditions. **a** Domain structure and phosphorylation sites of the AGC family kinase Sch9. TORC1 targets residues within the C-terminal domain (CT): T723 and S726 are located in a region, which resembles the turn motif (TM) of AGC kinases, T737 is located in the conserved hydrophobic motif (HM), which serves as the docking site for PDK, S758 and S765 are located in HM-like regions[46, 49]. These 5 TORC1-dependent sites were changed to alanines in the Sch9-5A mutant or to glutamate/aspartate in the Sch9-2D3E mutant[24]. PDK targets residue T570 within the activation loop[24]. SNF1 targets poorly characterized residues within a region of unknown function[60, 61]. **b–e** Glucose-grown Δsch9 cells expressing Sch9-FLAG, Sch9-5A-FLAG, Sch9-T570A-FLAG or Sch9-T570A/5A-FLAG were treated with cycloheximide (+CHX), rapamycin (+rapa), or were starved 10 min for glucose as indicated. Total protein extracts were resolved on phos-tag or Tris-Tricine (PAGE) gels and were subsequently analyzed via immunoblotting with α-FLAG or α-phospho-T570 (α-T570-Pi) antibodies. If indicated, total extract was treated with lambda phosphatase (λP) prior to the analysis. Sch9 species prominent on glucose (S0, S1, S3) are indicated in black, species prominent after starvation (S2, S4, S2**) are indicated in blue. **f** Cartoon summary of Sch9 phosphorylation. In glucose-rich conditions Sch9 is distributed between S0, S1, and S3. In glucose-starved conditions Sch9 is distributed between S4 and S2. Sch9-T570A/5 A is non-phosphorylated in glucose-rich conditions (S0) and becomes phosphorylated at a single residue upon starvation (S2**). The kinases responsible for the respective phosphorylation events are indicated. PDK-dependent phosphorylation, *blue circle*; TORC1-dependent phosphorylation; *red circles*; SNF1-dependent phosphorylation, *yellow circle*; residues mutated to alanines are indicated by *red asterisks*. For more details compare Results section

Indeed, upon glucose depletion the bulk of Sch9-T570A/5 A became phosphorylated to a species termed Sch9-S2** (S2**) (Fig. 1e). S2** and S1 co-migrated on phos-tag gels suggesting that both represented Sch9 species phosphorylated at a single residue.

The AMP-activated protein kinase (AMPK) homolog SNF1 is activated quickly upon glucose depletion and counteracts the TORC1 pathway[21, 25]. We thus tested the possibility that SNF1 was responsible for the phosphorylation of Sch9 upon glucose withdrawal. In Δsnf1 cells grown in glucose-rich conditions Sch9 phosphorylation resembled the wild type, however, when Δsnf1 cells were glucose depleted, S2 and S4 did not appear and S3 remained stable (Fig. 1e). Moreover, in Δsnf1 cells Sch9-5A S2* (Supplementary Fig. 1a and Supplementary Note 1) and Sch9-T570A/5 A S2** (Fig. 1e) were not formed upon glucose depletion. Thus, Sch9 was phosphorylated in an SNF1-dependent manner quickly upon glucose removal from the medium (Fig. 1f, right panel and Supplementary Fig. 1c). The findings reveal that non-phosphorylated Sch9 as well as species phosphorylated by PDK and/or TORC1 target sites were

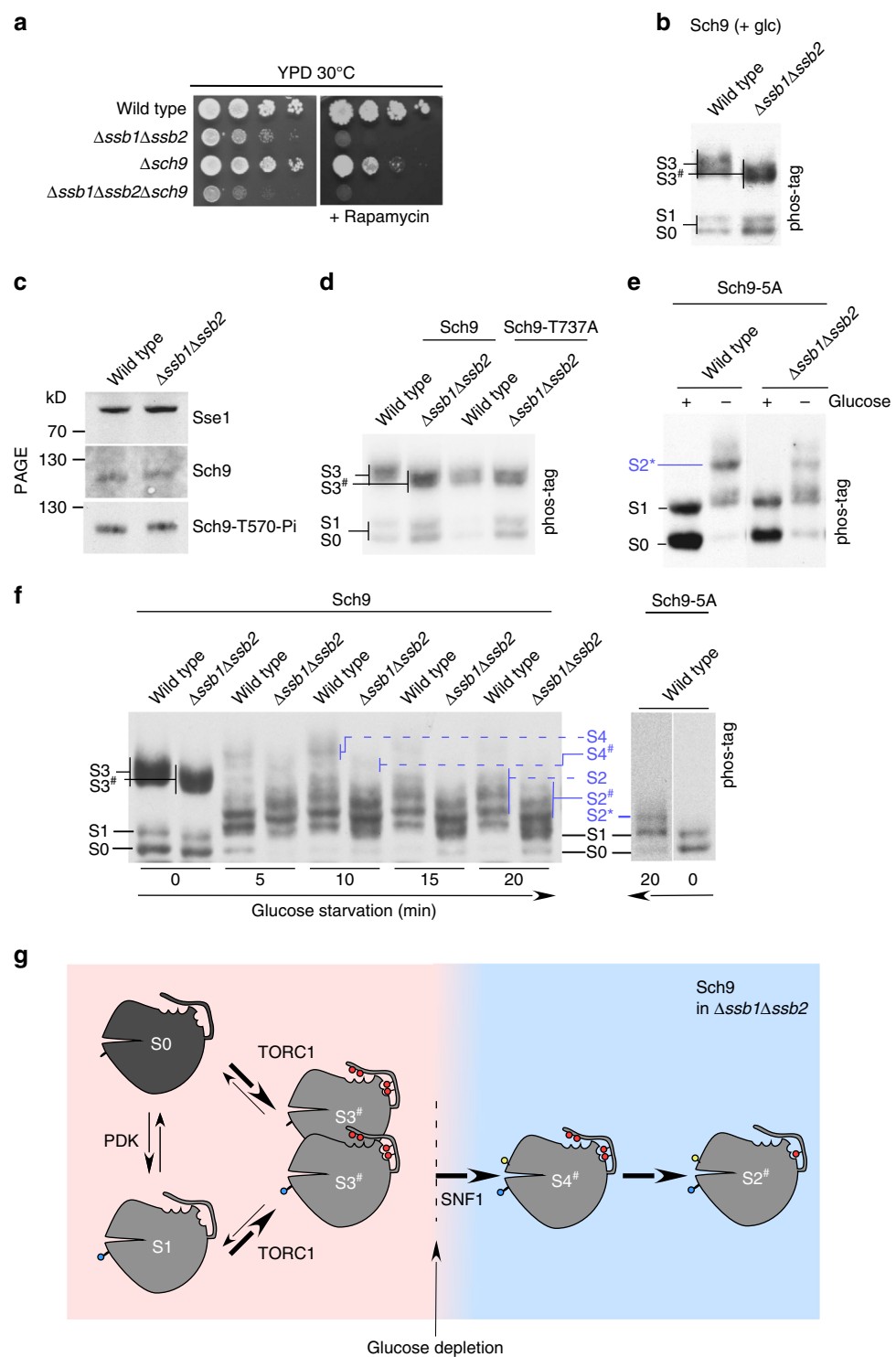

phosphorylated in an SNF1-dependent manner upon glucose depletion (Fig. 1c, e). Because SNF1 can directly phosphorylate Sch9 in vitro[30] the data strongly suggested that phosphorylation of Sch9 was directly by SNF1 (Fig. 1f and see also Discussion). Of note, changes with respect to Sch9 phosphorylation upon glucose depletion were distinct from those upon nitrogen starvation. In the latter case the bulk of Sch9 became quickly dephosphorylated at all of the TORC1-dependent sites, resulting in the accumulation of the S0/S1 species (Supplementary Fig. 1b). SNF1-dependent phosphorylation was not observed upon nitrogen starvation (Supplementary Fig. 1b).

**Ssb is required for phosphorylation of Sch9 on Thr-737.** Strains lacking Sch9 or Ssb resemble each other with respect to ribosome biogenesis defects. While the underlying principles have been unraveled in the case of Sch9 signaling, the role of Ssb in ribosome biogenesis is only poorly understood (see Introduction). We found that cells lacking Ssb displayed sensitivity towards rapamycin, which is indicative of alterations in the TORC1 pathway (Fig. 2a). We thus tested the possibility that Ssb might affect ribosome biogenesis via the TORC1-Sch9 pathway. Phos-tag gel analysis revealed that in glucose-grown cells lacking Ssb, Sch9 was distributed between S0 and S1, and a faster migrating form of S3, termed Sch9-S3# (S3#) (Fig. 2b). PDK-dependent phosphorylation of Sch9 in cells lacking Ssb was unaffected, as indicated by i) a similar distribution of Sch9 between the S0 and S1 species (Fig. 2b) and ii) a similar phosphorylation level of Sch9-T570 (Fig. 2c). The data suggested that in the absence of Ssb one, or more, of the 5 residues phosphorylated by TORC1 were erroneously lacking phosphorylation.

Of the 5 TORC1-dependent residues in the Sch9-CT (Fig. 1a) only the T737A single mutation results in phenotypic defects (ref. [24] and Supplementary Fig. 2a). We thus compared the phosphorylation pattern of wild-type Sch9 expressed in a Δssb1Δssb2 background with the phosphorylation pattern of the Sch9-T737A mutant. Indeed, the S3# species formed in Δssb1Δssb2 cells co-migrated with the Sch9-T737A mutant independent of whether Sch9-T737A was expressed in the wild-type or a Δssb1Δssb2 background (Fig. 2d). The data are consistent with a model, in which Sch9 lacked phosphorylation at T737 when Ssb was absent. To test if Ssb also affected SNF1-dependent phosphorylation, we employed the Sch9-5A mutant (Fig. 2e). However, this was not the case. Upon glucose depletion SNF1-dependent phosphorylation of Sch9-5A to S2* occurred in wild-type (Fig. 2e and Supplementary Fig. 1a) as well as in Δssb1Δssb2 cells (Fig. 2e).

To get more insight into the dynamics of Sch9 phosphorylation upon glucose depletion we performed a time course (Fig. 2f). The analysis revealed complex changes of Sch9 phosphorylation. In wild-type cells, S4 was formed transiently, with a peak after approximately 10 min, and a significant down-shift to the S2 species after about 15–20 min of glucose depletion (Fig. 2f). In Δssb1Δssb2 cells, the changes of the Sch9 phosphorylation pattern were similar, however, the species corresponding to S2 and S4 migrated faster, forming S2# and S4# (Fig. 2f, g). The most simple explanation for the distinct phosphorylation patterns is that upon glucose depletion Sch9 remained phosphorylated at T737 in wild-type cells, while in Δssb1Δssb2 cells Sch9 was lacking T737 phosphorylation to start with, resulting in faster migrating Sch9 species throughout the time course (Figs. 1f, 2f, g). The combined data indicate that Ssb was required for proper TORC1-dependent phosphorylation of Sch9-T737 (Fig. 2g, compare Fig. 1f).

**Ssb binds to Sch9 in an ATP-sensitive manner.** In order to test if Ssb affected Sch9 via direct interaction, we performed co-immunoprecipitation experiments with Sch9-FLAG. As a control, we employed Maf1, a transcriptional repressor, which is a well characterized direct Sch9 target[18, 26, 27]. Indeed, a low amount of Maf1 was co-immunoprecipitated with Sch9-FLAG (Fig. 3a). In the same reaction also Ssb was co-immunoprecipitated with Sch9-FLAG, indicating that Ssb was bound to Sch9 either directly, or Ssb and Sch9 were part of a larger complex (Fig. 3a). Consistent with the typical interaction of Hsp70 homologs with their substrate proteins, which is weakened when Hsp70s are in the ATP-bound state[1], Ssb was released from Sch9-FLAG when ATP was added to the reaction (Fig. 3b). On the basis of copy-number estimates of yeast proteins[31] and a side by side comparison of the material bound to FLAG-beads to serial dilutions of total protein extracts, we estimated that roughly 4% of cellular Sch9-FLAG was in a complex with Ssb, while only 0.0025% was in a complex with Maf1 (Supplementary Fig. 3a, b). While this is a rough estimate, the very low amount of Maf1 bound to Sch9-FLAG is consistent with the transient kinase/substrate interaction. The finding that a much larger fraction of Sch9-FLAG was bound to Ssb suggests that Sch9 was a client of Ssb.

**Sch9 activity towards Maf1 requires Ssb.** As expected, in glucose-grown wild-type cells Maf1 was hyper-phosphorylated, while in Δsch9 cells Maf1 was largely non-phosphorylated (Fig. 3c, d), a condition known to inhibit Pol III[18, 26, 27]. Consistent with the inactivation of Sch9, Maf1 phosphorylation was strongly reduced after glucose depletion (Fig. 3d). The dephosphorylation of Maf1 was dependent on SNF1, which is consistent

**Fig. 2** Ssb is required for proper TORC1-dependent phosphorylation of Sch9-T737. **a** Cells lacking Ssb display rapamycin sensitivity. Serial dilutions of logarithmically growing cells were spotted onto YPD plates with or without 6 ng ml⁻¹ rapamycin and were incubated at 30 °C. **b** Sch9 is hypo-phosphorylated in the absence of Ssb. Protein extracts from glucose-grown Δsch9 cells expressing Sch9-FLAG were analyzed on phos-tag gels as described in Fig. 1. S3# indicates the faster migrating species detected in Δssb1Δssb2. **c** T570 phosphorylation is unaffected by Ssb. Total extracts of the strains indicated were resolved on Tris-Tricine gels and were subsequently analyzed via immunoblotting with α-T570-Pi, α-Sch9, and α-Sse1 (loading control). **d** Sch9 phosphorylation at T737 depends on Ssb. Protein extracts from Δsch9 cells expressing Sch9-FLAG or Sch9-T737A-FLAG were analyzed on phos-tag gels as described in Fig. 1. S3# indicates the faster migrating form of S3 seen in the absence of Ssb, which comigrates with the Sch9-T737A S3# species. **e** SNF1-dependent phosphorylation is unaffected by Ssb. Protein extracts from Δsch9 cells expressing Sch9-5A-FLAG in glucose-rich or glucose-starved (10 min) conditions were analyzed on phos-tag gels as described in Fig. 1. For analysis of the Sch9-5A mutant the concentration of the phos-tag reagent was increased for better resolution (Methods section). **f** Sch9 is hypo-phosphorylated at T737 in Δssb1Δssb2 in the presence or absence of glucose. Glucose-grown Δsch9 cells expressing Sch9-FLAG (left panel) or Sch9-5A-FLAG (right panel) were depleted for glucose for the indicated times and were analyzed on phos-tag gels as described in Fig. 1. Sch9 species, which migrated faster in Δssb1Δssb2 cells when compared to the wild type are indicated with #. **g** Cartoon summary of Sch9 phosphorylation in Δssb1Δssb2 cells. In glucose-rich conditions Sch9 is distributed between S0, S1, and S3# in Δssb1Δssb2. When compared to S3 (Fig. 1f) S3# lacks the phosphorylation at residue T737. In glucose-starved conditions Sch9 is distributed between S2#, and S4#, which also lack phosphorylation at T737 when compared to the corresponding S2 and S4 species seen in wild type (Fig. 1f). For more details compare Results section

with the observation that upon glucose depletion Sch9 remained phosphorylated at the PDK and TORC1 residues in the absence of SNF1 (Fig. 1e and Supplementary Fig. 3c, d). In $\Delta ssb1\Delta ssb2$ cells Maf1 phosphorylation was severely reduced (Fig. 3d). Of note, Maf1 and Sch9 protein levels were not affected in cells lacking Ssb (Fig. 3e) and the Maf1 phosphorylation defect was not due to the partial activation of SNF1, which occurs in cells lacking Ssb (Supplementary Fig. 3c and Supplementary Note 2)[9, 10].

Consistent with the importance of Sch9-T737 phosphorylation, Maf1 was non-phosphorylated in cells expressing Sch9-T737A (Fig. 3f). Thus, when Ssb was absent Sch9 was not only improperly phosphorylated (Fig. 2), but was also less efficient in phosphorylating Maf1. The requirement for Ssb with respect to Sch9 activation, however, was not absolute, as overexpression of Sch9 in the $\Delta ssb1\Delta ssb2$ background partially restored hyperphosphorylation of Maf1 (Fig. 3d).

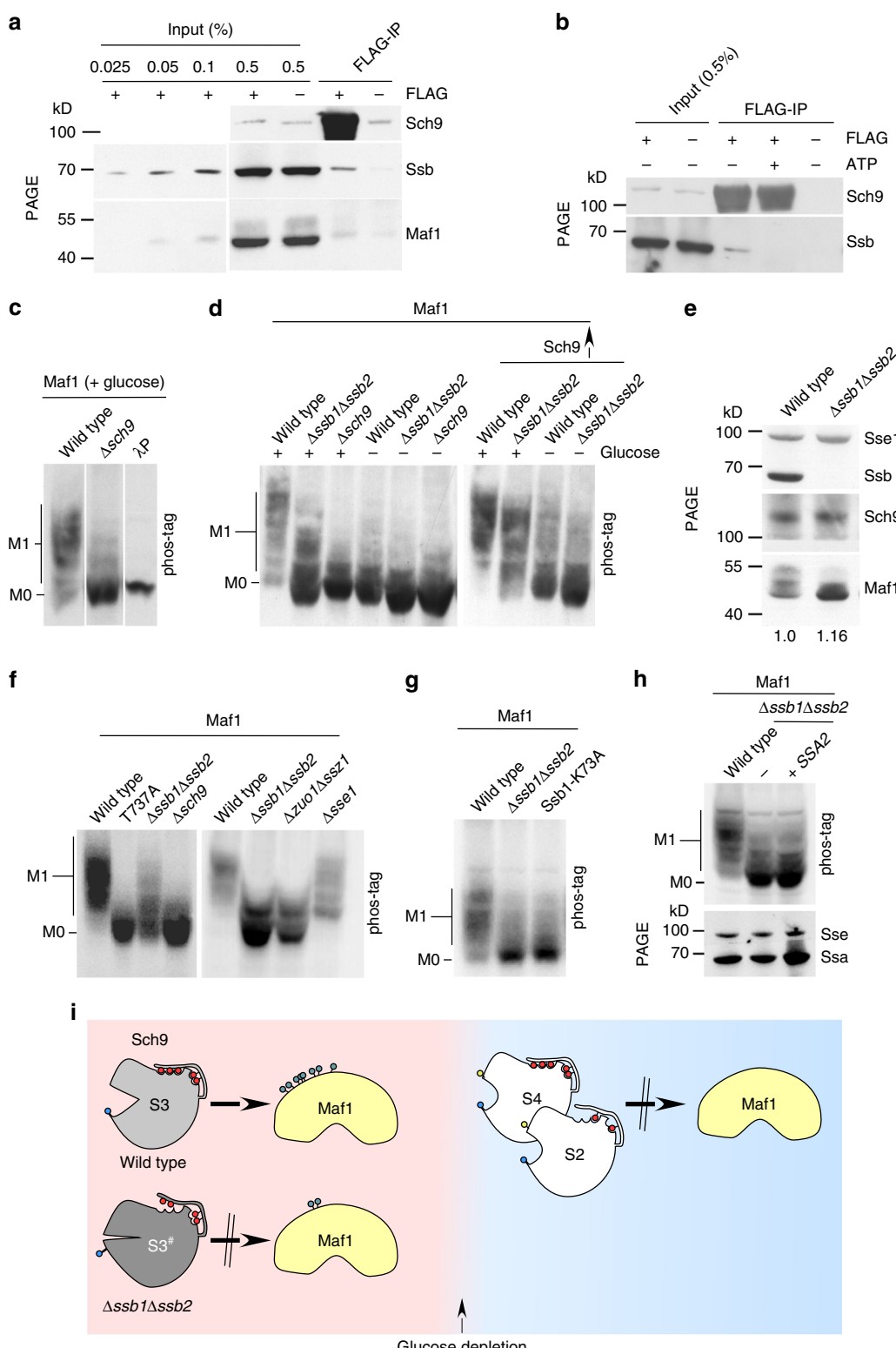

We next employed Maf1 as a reporter to determine the effect of other cytosolic chaperones with respect to Sch9 activation. First, we tested Maf1 phosphorylation in the absence of the Ssb cochaperone RAC[7]. Cells lacking RAC displayed a strong Maf1 phosphorylation defect (Fig. 3f), indicating that Ssb´s ATPase-driven cycle of substrate binding and release was required for proper activation of Sch9. Consistently, Maf1 was hypo-phosphorylated also in cells expressing the Ssb-K73A mutant[32], which cannot hydrolyze ATP (Fig. 3g). Second, the nucleotide exchange factor Sse1 did not significantly affect Maf1 phosphorylation (Fig. 3f). Sse1 interacts with Ssa as well as Ssb, however, based on currently available data, Sse1 predominantly functions as a cochaperone of Ssa in the cytosol[2]. The data therefore suggested that the Ssa/Sse1 chaperone couple was not required for Sch9 activation. Third, overexpression of Ssa in a Δssb1Δssb2 background did not restore Maf1 phosphorylation (Fig. 3h). The combined data support a model in that Ssb, together with its cochaperone RAC, serves a specific function with respect to Sch9 activation (Fig. 3i), which cannot be substituted by Ssa, even though Ssa is present in Sch9-FLAG isolates (Supplementary Fig. 3e).

**Sch9 and Ssb functions in ribosome biogenesis overlap.** Because Sch9-dependent phosphorylation of Maf1 is one of the requirements for proper ribosome biogenesis[18, 26, 27] we speculated that ribosome biogenesis in cells lacking either Ssb or Sch9 was affected in a similar way. To test this hypothesis we made use of the observation that ribosomal particles are recovered in aggregates in cells lacking Ssb[12] and tested if aggregation of ribosomal particles also occurred in cells lacking Sch9. This was indeed the case, aggregation of ribosomal particles occurred in Δssb1Δssb2, Δsch9, and Δssb1Δssb2Δsch9, while aggregation was significantly lower in wild-type cell extracts (Fig. 4a and b). Immunoblotting revealed that with respect to the aggregation of small ribosomal subunit proteins, SSB1/SSB2 and SCH9 displayed strong genetic interaction such that the extent of aggregation was similar in Δssb1Δssb2 and Δsch9 cells, and was not further enhanced in Δssb1Δssb2Δsch9 cells (Fig. 4b). Interestingly, aggregation of large ribosomal subunit proteins was seemingly affected differentially by Ssb and Sch9. With respect to the large subunit proteins tested, aggregation was more pronounced in Δssb1Δssb2 and Δssb1Δssb2Δsch9 cells when compared to Δsch9 cells (Fig. 4b). The analysis suggested that the effect of Ssb with respect to small ribosomal subunit biogenesis was mainly via Sch9, while Ssb affected large ribosomal subunit biogenesis via Sch9 and, in addition, via a Sch9-independent pathway. To further test this model we analyzed a permanently active phosphomimetic of Sch9

(Sch9-2D3E), in which the residues targeted by TORC1 were replaced by acidic residues[24] (Fig. 1a). Expression of Sch9-2D3E in the Δssb1Δssb2 background partially rescued ribosome aggregation (Fig. 4c, upper panel) and restored Maf1 phosphorylation (Fig. 4d). Consistent with a differential affect of Ssb with respect to small vs. large ribosomal subunit biogenesis, aggregation of a small subunit protein was efficiently suppressed, while aggregation of a large subunit protein was only partly suppressed when Sch9-2D3E was expressed in Δssb1Δssb2 cells (Fig. 4c, lower panel). As expected based on the multiple functions of the Ssb/RAC system[2, 7], Sch9-2D3E expression was not sufficient to complement the general growth defects of cells lacking Ssb (Fig. 4e).

**Basal phosphorylation of Hsf1 depends on Sch9 and Ssb.** The above data suggested that Ssb affected TORC1-Sch9 signaling and by that affected ribosome biogenesis. However, Ssb also binds cotranslationally to ribosomal proteins and prevents their misfolding[12–14]. We thus wanted to determine if the misfolding of ribosomal proteins (and also other) newly synthesized proteins was increased to a level, which could significantly contribute to the severely reduced concentration of ribosomes and to ribosome aggregation in cells lacking Ssb[2, 12, 14].

To answer this question we employed Hsf1 as a reporter, because Hsf1 is activated when the concentration of misfolded proteins in a cell rises, for example upon heat shock, or when the toxic proline analog, L-azetidine-2-carboxylic acid (AZC) is incorporated into newly synthesized polypeptides[33–35]. Hsf1 activation upon protein misfolding is regulated via a negative feedback loop, which involves the chaperones Hsp90, Ssa, and Sse1[34, 35]. Misfolded polypeptides increase the substrate-load of Ssa and Hsp90 and this titrates the chaperones away from Hsf1, resulting in phosphorylation and activation of Hsf1[34–36]. Hsf1 is also activated during the shift from fermentation to respiration, or when glucose levels drop[37–39]. The mechanism which leads to the activation of Hsf1 upon glucose depletion is not fully understood, however, it is distinct from that upon heat shock and involves phosphorylation of Hsf1 by the kinase Yak1[34, 39].

In glucose-grown wild-type cells, Hsf1 migrated as a doublet termed Hsf1-H1/H2 (H1/H2) on phos-tag gels (Fig. 5a and Supplementary Fig. 4a, b). Upon glucose depletion H1/H2 shifted to more highly phosphorylated species termed Hsf1–H3 (H3) (Fig. 5a, Supplementary Fig. 4c). On the basis of our hypothesis (see above) we tested if Hsf1 was hyper-phosphorylated when Ssb was absent, as expected if unfolded or misfolded proteins accumulated to a significant extent. However, this was not the case. Rather, in glucose-grown cells

**Fig. 3** Sch9 kinase activity is disturbed in cells lacking Ssb. **a, b** FLAG-IP reactions were performed with extracts derived from Δsch9 cells overexpressing either Sch9 (−FLAG) or Sch9-FLAG (+FLAG). If indicated ATP was added at a concentration of 10 mM. Total extracts (input) and immunoprecipitated proteins (FLAG-IP) were separated on Tris-Tricine gels and were subsequently analyzed via immunoblotting using α-Sch9, α-Ssb, and α-Maf1. Details are described in Methods section and in Supplementary Fig. 3a, b. **c, d** Maf1 phosphorylation is reduced in Δssb1Δssb2 cells. Strains as indicated were grown on glucose (+glucose) or were depleted for glucose for 10 min (−glucose) prior to collection. Strains labeled with Sch9↑ overexpress Sch9. Protein extracts were analyzed via phos-tag gels followed by immunoblotting with α-Maf1. M0: dephosphorylated Maf1 species, M1: hyper-phosphorylated Maf1 species. λP: lambda phosphatase treated extract. **e** Protein levels of Sch9 and Maf1 are unaffected in cells lacking Ssb. Extracts from glucose-grown strains were resolved on a Tris-Tricine gel and were subsequently analyzed via immunoblotting with α-Maf1, α-Sse1, α-Ssb, or α-Sch9. The numbers below the Maf1 bands indicate the intensity of the total Maf1 signal in Δssb1Δssb2 cells relative to that in wild-type cells. **f** Maf1 hyper-phosphorylation requires phosphorylation of Sch9-T737 and the Ssb co-chaperone RAC. Protein extracts from indicated strain were analyzed via phos-tag gels followed by immunoblotting with α-Maf1. **g** Maf1 phosphorylation requires ATP-hydrolysis by Ssb. Analysis as in f. **h** Overexpression of Ssa does not restore Maf1 hyper-phosphorylation in cells lacking Ssb. Aliquots of total cell extracts as indicated were separated via phos-tag gels (upper panel) or Tris-Tricine gels (lower panel) and were analyzed via immunoblotting with α-Maf1, α-Ssa, or α-Sse1 (loading control). **i** Cartoon summary of Maf1 phosphorylation in wild-type and Δssb1Δssb2 cells. In glucose-rich conditions Maf1 is hyper-phosphorylated by Sch9-S3, while Sch9-S2 and Sch9-S4, formed upon glucose depletion are inactive towards Maf1. Sch9-S3# in Δssb1Δssb2 cells, lacking phosphorylation at T737, is not fully active with respect to Maf1 phosphorylation. For more details compare Results section

lacking Ssb, Hsf1 migrated slightly faster when compared to the wild type and formed the hypo-phosphorylated species termed Hsf1-H1*/H2* (H1*/H2*) (Fig. 5a and Supplementary Fig. 4c).

During the analysis of Hsf1 we also found that upon glucose depletion in cells lacking Ssb, the H1/H2 species were shifted up only slightly to a species termed Hsf1–H3* (H3*) while H3 was not formed (Fig. 5a and Supplementary Fig. 4c). Because of the

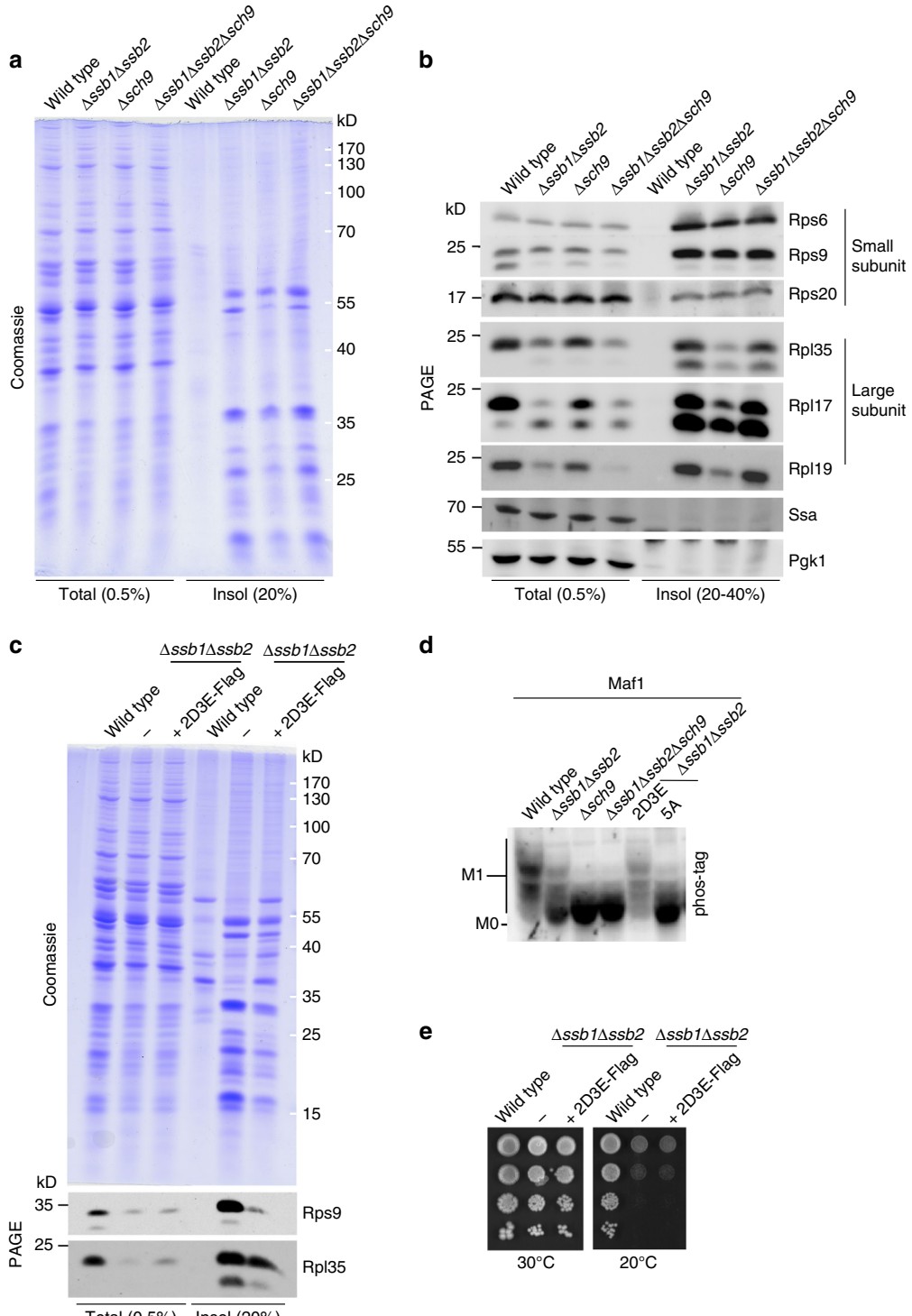

**Fig. 4** Ribosome aggregation in cells lacking Ssb is due to reduced Sch9 activity. **a, b** Aggregation of ribosomal proteins in cells lacking Ssb and/or Sch9. Insoluble material was prepared as described in Methods section. The indicated fractions of total cell extract (total) and insoluble material (insol) were separated on Tris-Tricine gels and were subsequently analyzed via Coomassie staining (**a**) or immunoblotting (**b**) employing the antibodies indicated. **c** Hyperactive Sch9-2D3E partially rescues ribosome aggregation in cells lacking Ssb. The experiment was performed and analyzed as described in **a, b**. **d** Hyper-active Sch9-2D3E partially rescues Maf1 phosphorylation in cells lacking Ssb. Protein extracts from the indicated strains were analyzed as described in Fig. 3c. **e** Hyper-active Sch9-2D3E does not complement for the general growth defects of cells lacking Ssb. Logarithmically growing cells from the strains indicated were serially diluted, spotted on YPD plates, and were incubated at the indicated temperatures

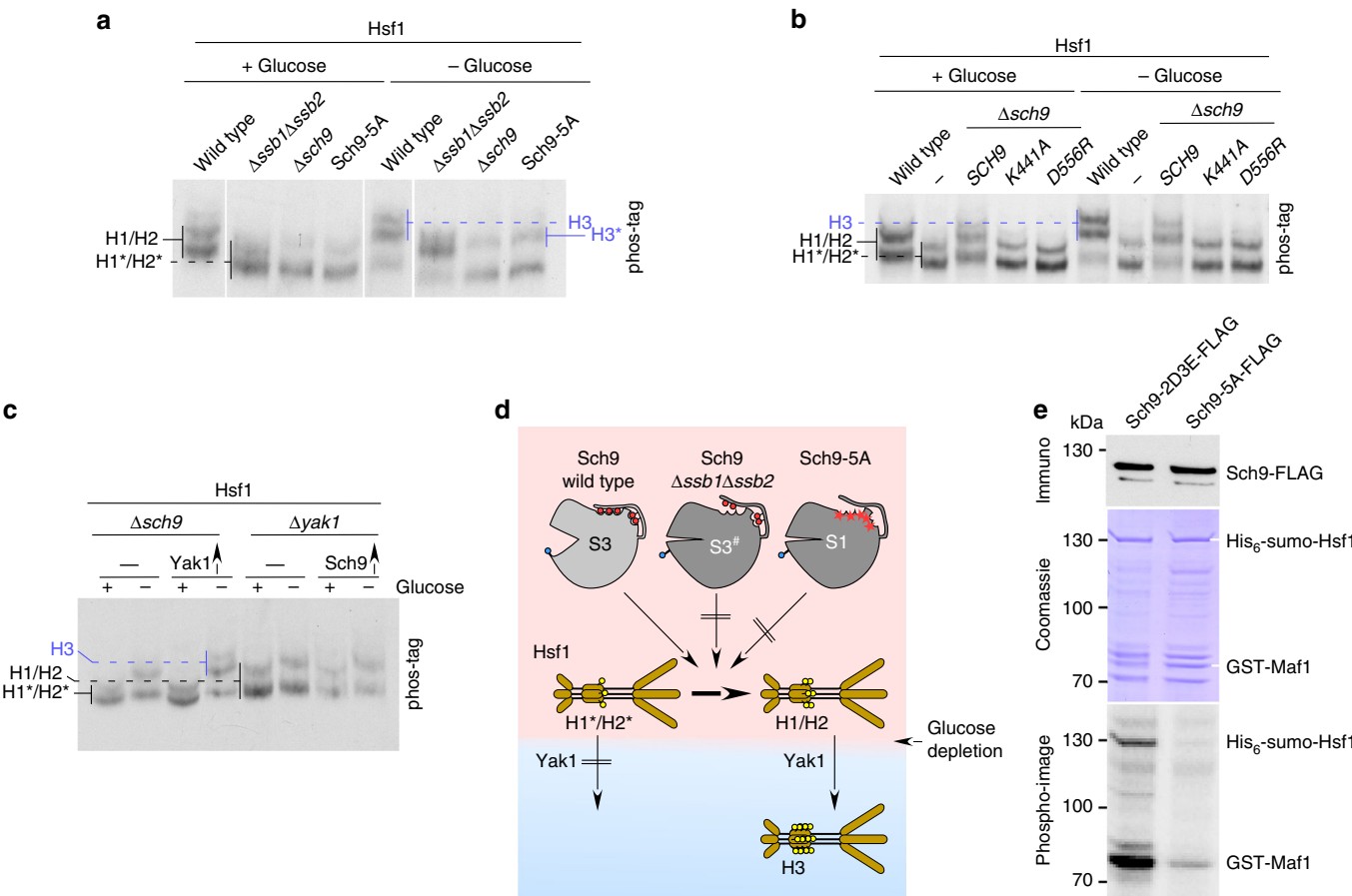

**Fig. 5** Sch9 phosphorylates Hsf1 in glucose-rich conditions. **a**–**c** Hsf1 phosphorylation is affected by the absence of Ssb or functional Sch9. Strains as indicated were grown on glucose (+glucose) or were glucose depleted for 10 min (−glucose) prior to collection. Protein extracts were analyzed via phos-tag gels followed by immunoblotting with α-Hsf1. H1/H2: predominant Hsf1 species in glucose-grown wild-type (**a**, **b**) and Δyak1 (**c**) cells; H1*/H2*: faster migrating species in Δssb1Δssb2 cells, Δsch9 cells, or in cells expressing non-functional Sch9 versions as indicated. H3: predominant hyper-phosphorylated Hsf1 species formed upon glucose depletion. H3*: hypo-phosphorylated Sch9 species form in cells lacking Ssb or active Sch9 upon glucose depletion. Arrows indicate overexpression of Sch9 or Yak1, respectively. **d** Cartoon summary of Hsf1 phosphorylation in glucose-rich and glucose-starved conditions. In glucose-grown wild-type cells Hsf1 is phosphorylated to H1/H2. In glucose-grown Δssb1Δssb2 cells, or in cells lacking functional Sch9, Hsf1 is hypo-phosphorylated to H1*/H2*. Yak1, activated upon glucose depletion, efficiently phosphorylates H1/H2 to H3, however, does not efficiently phosphorylate the hypo-phosphorylated species H1*/H2*. **e** Sch9 phosphorylates Hsf1 in vitro. A mixture of partially purified His$_6$-sumo-Hsf1 and GST-Maf1 was incubated with Sch9-2D3E-FLAG or Sch9-5A-FLAG in the presence of γ-$^{32}$P-ATP (see Methods section). The total amount Sch9-2D3E-FLAG or Sch9-5A-FLAG in the reaction was analyzed via immunoblotting with α-FLAG (*upper panel*), total proteins in the reactions were visualized by Coomassie staining (*middle panel*, and see also Supplementary Fig. 5), incorporation of $^{32}$P was analyzed via exposure of the Coomassie stained gel to a phosphor image plate (*lower panel*)

effect of Ssb on Sch9 activity (Figs 2–4) we tested if Hsf1 phosphorylation in glucose-rich conditions was affected by Sch9. This was indeed the case, in glucose-grown cells lacking Sch9 or expressing Sch9-5A, Hsf1 was hypo-phosphorylated like in cells lacking Ssb (Fig. 5a). Moreover, phosphorylation of Hsf1 to H1/H2 in the Δssb1Δssb2 background was restored upon expression of the Sch9-2D3E hyperactive mutant (Supplementary Fig. 4d). Upon glucose depletion species comigrating with H3*, but not H3, were formed in Δsch9 or Sch9-5A cells (Fig. 5a, and Supplementary Fig. 4e). Confirming this result, Δsch9 cells expressing the enzymatically inactive mutants Sch9-K441A or Sch9-D556R[40] showed the same defect in Hsf1 phosphorylation (Fig. 5b). These observations strongly suggested that the Hsf1 phosphorylation defect in the absence of Ssb was connected to the Sch9 phosphorylation defects described above.

The kinase Yak1 phosphorylates Hsf1 upon brief glucose depletion[39]. Confirming this, in Δyak1 cells Hsf1–H3 formation was abolished when glucose was depleted (Fig. 5c and Supplementary Fig. 4f). Because active Sch9 was not only

required for the proper phosphorylation of Hsf1 in glucose-rich conditions, but also for the subsequent hyper-phosphorylation upon glucose depletion, we speculated that Yak1 preferred H1/H2 over H1*/H2* as a substrate. Indeed, Yak1 overexpression in Δsch9 cells failed to rescue the Hsf1 phosphorylation defect in glucose-rich conditions, however, restored Hsf1 hyper-phosphorylation upon glucose depletion (Fig. 5c). In contrast, Sch9 overexpression in Δyak1 cells did not restore H3 formation (Fig. 5c). The combined data suggested that Sch9 was involved in a priming phosphorylation of Hsf1 to H1/H2 in glucose-rich conditions, which facilitated the Yak1-dependent hyper-phosphorylation to H3 that occurred upon glucose depletion (Fig. 5d, and Discussion section).

**Hsf1 is a substrate of Sch9 kinase in vitro.** In order to test the possibility that Hsf1 was a direct target of Sch9, we performed in vitro kinase assays with partially purified His$_6$-sumo-Hsf1 and, as a control GST-Maf1[41] (Supplementary Fig. 5). FLAG-beads

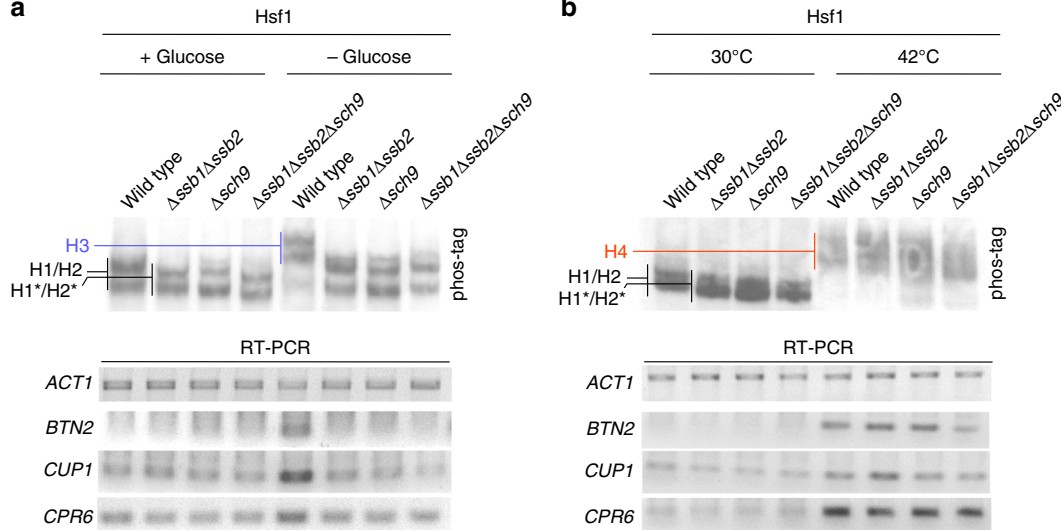

**Fig. 6** Phosphorylation of Hsf1 by Sch9 is required for Hsf1 activation upon glucose depletion but is not required for Hsf1 activation upon heat shock. **a**, **b** Total protein or total RNA was prepared from glucose-grown, glucose depleted, or heat shocked (42 °C, 15 min) cells as described in Methods section. Analysis of Hsf1 phosphorylation was performed as described in Fig. 5. H4: hyper-phosphorylated Hsf1 species formed upon heat shock. RT–PCR for the genes indicated was performed with total RNA. Actin (ACT1) levels were used as a loading control. Glucose starvation in RT–PCR experiments was 30 min to allow for the expression of a detectable amount of Hsf1-induced transcripts. The Hsf1–H3 species was stable during a 30 min glucose depletion time course (Supplementary Fig. 4c)

coated with either permanently active Sch9-2D3E-FLAG or with inactive Sch9-5A-FLAG were incubated with a mixture of His$_6$-sumo-Hsf1 and GST-Maf1 (Fig. 5e and Supplementary Fig. 5) in the presence of γ-$^{32}$P-ATP. As expected, GST-Maf1 was phosphorylated by Sch9-2D3E-FLAG, but not by Sch9-5A-FLAG (Fig. 5e). In the same reaction mix, His$_6$-sumo-Hsf1 was phosphorylated by Sch9-2D3E-FLAG, but not by Sch9-5A-FLAG (Fig. 5e). We conclude that Hsf1 was a direct target of Sch9 in vitro.

**Hsf1 activation upon glucose depletion requires Hsf1-H1/H2**. To test if Sch9 and Ssb were required for gene induction upon brief glucose depletion, we analyzed the expression of the Hsf1 target genes BTN2, CUP1, and CPR6[37, 42, 43], employing ACT1 as a control. In the wild-type strain transcript levels of BTN2, CUP1, and CPR6 were increased upon glucose depletion (Fig. 6). As shown above, glucose depletion in cells lacking Sch9 and/or Ssb did not result in hyper-phosphorylation of Hsf1 and consistently the transcript levels of BTN2, CUP1, and CPR6 remained low (Fig. 6a). Thus, Sch9 and Ssb were not only required for proper phosphorylation of Hsf1, but also for Hsf1-dependent target gene induction upon glucose depletion. Importantly, Sch9 and Ssb were not required for the activation of Hsf1 upon heat shock. Hsf1 was strongly hyper-phosphorylated upon heat shock, and the phosphorylation pattern was significantly different from that observed upon glucose depletion (Fig. 6 and Supplementary Fig. 4b)[38]. Heat shock induced Hsf1 hyper-phosphorylation was not significantly affected in cells lacking Sch9 and/or Ssb, indicating that the phosphorylation that occurred upon heat shock was independent of the Sch9 priming phosphorylation (Fig. 6b). Consistently, upon heat shock induction of Hsf1 target genes was similar in wild type and strains lacking Sch9 and/or Ssb (Fig. 6b). The findings indicate that in cells lacking Ssb Hsf1 was not activated due to the accumulation of unfolded proteins, however, Hsf1 became readily activated when unfolded proteins accumulated upon heat shock. Moreover, the ability of Hsf1 to become activated upon heat shock

confirmed that Ssb was not required for the de novo folding of Hsf1, or to keep Hsf1 properly folded.

**Discussion**

Our data reveal a novel function of Ssb, which is essential for the TORC1-dependent activation of the kinase Sch9. Because Sch9 controls ribosome biogenesis in unstressed conditions, our findings provide a mechanistic explanation for the ribosome biogenesis defects in cells lacking Ssb. Moreover, we show that in the absence of Ssb the load of unfolded or aggregated proteins was not enhanced to a level, which activated the stress responsive transcription factor Hsf1, likely, because the folding function of Ssb was taken over by other folding chaperones, in particular by Ssa. The essential role in TORC1-Sch9 signaling, however, was confined to Ssb. Several major conclusions are supported by the data in this study.

First, we defined the major phosphorylated species of Sch9 in glucose-rich and glucose-deplete conditions employing a combination of yeast mutants, phospho-specific antibodies, and drugs (Fig. 1). The phos-tag method was instrumental in this, because separation of Sch9 on these gels allowed us to distinguish between the differently phosphorylated species of full length Sch9, without the need of 2-nitro-5-thiocyanatobenzoic acid (NTCB) induced cleavage[24, 25]. In addition, it allowed us to assess the relative distribution of Sch9 between its differently phosphorylated forms. In glucose-grown wild-type cells, about half of Sch9 was phosphorylated at T570 by PDK and more than 75% of Sch9 was phosphorylated by TORC1[24]. Beyond that, the phos-tag gel analysis revealed that upon glucose depletion, but not upon nitrogen depletion, a large fraction of Sch9 became phosphorylated in an SNF1-dependent manner.

In mammalian cells, AMPK is upstream of TORC1 and suppresses TORC1 signaling[44], however, in yeast this is not firmly established. It was recently shown that in the absence of SNF1, the TORC1-dependent residues in the Sch9-CT remain phosphorylated upon glucose depletion and that SNF1 was required to switch off TORC1-Sch9 signaling upon glucose

depletion[25]. One possible explanation for this observation is that SNF1 phosphorylates and inactivates TORC1 itself, as found in mammalian cells[44]. Our data suggest that SNF1-dependent phosphorylation of Sch9 itself may also contribute to dephosphorylation of the TORC1-dependent residues within the Sch9-CT, for example by recruitment of an as yet unidentified Sch9 phosphatase.

Second, we show that Ssb is required for the proper phosphorylation of Sch9 and interacts with Sch9 in an ATP-sensitive manner (Figs 2 and 3). We find that efficient TORC1-dependent phosphorylation of specifically Sch9-T737, which is essential for Sch9 activity[24] depends on Ssb, while the other TORC1-dependent residues are phosphorylated in an Ssb-independent manner. How could Ssb affect the phosphorylation of Sch9? The interaction of Ssb with Sch9 follows the general principles of Hsp70 function (Fig. 3). However, our data suggest that Ssb is not required for the overall folding of Sch9, but to facilitate TORC1-dependent phosphorylation of Sch9-T737. In this context it is interesting that recent findings revealed that Hsp70s not only interact with unfolded polypeptide segments but also with near-native conformations of substrate proteins[45]. Of note, the regulation of the AGC family kinase PKC depends on direct interaction with Hsp70[46]. The dephosphorylated, but not the phosphorylated, turn motif of PKC provides a specific binding site for Hsp70 and the binding of Hsp70 stabilizes PKC to allow for subsequent rephosphorylation[46–48]. As a consequence, disruption of Hsp70 binding leads to accelerated dephosphorylation of PKC[48]. Our data suggest that a similar event may result in reduced phosphorylation of Sch9-T737 in the absence of Ssb. Such motif-specific binding could also explain why only T737, but no other TORC1-dependent residue lacked proper phosphorylation in the $\Delta ssb1\Delta ssb2$ mutant. Interestingly, T737 is surrounded by hydrophobic residues within the so-called hydrophobic motif of Sch9 (Fig. 1a)[49]. Such hydrophobic segments are preferred binding sites of Hsp70 homologs[1]. Of note, we recently found that Ssb plays a similar posttranslational role with respect to the Glc7/Reg1-dependent dephosphorylation of SNF1[7, 10, 11].

Third, our data reveal that late steps of ribosome biogenesis were hampered in the absence of Ssb, because the TORC1-Sch9 signaling axis[18, 20, 50] was not fully active (Fig. 4). We show that in $\Delta ssb1\Delta ssb2$ cells Maf1 was not properly phosphorylated, which again results in the inhibition of Pol III-dependent transcription[18, 26, 27]. Pol III is responsible for the transcription of 5 S rRNA and also U6 spliceosomal RNA[26, 27]. 5 S rRNA is not only a component of the ribosome, but is also required for the processing of 27 S pre-rRNA[51]. Thus, reduced 5 S rRNA levels may provide an explanation for the strong accumulation of 27 S rRNA observed in cells lacking the Ssb/RAC system[14]. U6 RNA is required for the removal of introns, which are rare in yeast genes, however, are highly abundant in genes encoding for ribosomal proteins[52]. It is also interesting to recall that nuclear export of pre-60S complexes is inhibited upon rapamycin treatment leading to the nuclear accumulation of, e.g., Rpl25-GFP[50]. Of note, Rpl25-GFP also accumulates in the nucleus of $\Delta ssb1\Delta ssb2$ cells[12]. Inhibition of the TORC1-Sch9 signaling pathway thus provides an explanation for major ribosome assembly defects observed in the absence of the Ssb chaperone system. However, our data indicate that, Ssb affects ribosome biogenesis via additional Sch9-independent mechanisms, which seems to affect predominantly the large ribosomal subunit and may involve the folding function of Ssb.

Fourth, we show that Sch9 phosphorylates Hsf1 in vitro and in vivo (Fig. 5). Sch9-dependent phosphorylation of Hsf1 in glucose-rich conditions was required for the subsequent activating phosphorylation by Yak1 when glucose was depleted.

This is significant, because Hsf1 is basally active, maintaining transcription of selected genes, even under optimal growth conditions[43]. Microarray data show that the basal transcript levels of, e.g., BTN2 and CPR6 in glucose-grown cells are down-regulated to 44 and 68% in $\Delta sch9$ compared to wild type[28]. Interestingly, BTN2 and CPR6 are also down-regulated to 63 and 80% in $\Delta ssb1\Delta ssb2$ cells[9]. We conclude that it is very likely that Ssb affects Hsf1 function in vivo via the TORC1-Sch9 signaling pathway. Interestingly, it was also previously shown that Ssb has little effect with respect to Hsf1 activity during heat shock, however, fine-tunes Hsf1 function under normal growth conditions[53]. Moreover, Ssb — but not Ssa — interacts post-translationally with Hsf1[53] and by this may also affect the accessibility of Sch9-dependent residues within Hsf1.

If protein misfolding were a major problem in $\Delta ssb1\Delta ssb2$ cells, Hsf1, which is activated in response to the accumulation of unfolded, or misfolded proteins[33–35] should be phosphorylated and activated with respect to its downstream targets. However, contrary to expectations this was not the case. Hsf1 in $\Delta ssb1\Delta ssb2$ cells was hypo-phosphorylated in glucose-rich conditions and also failed to respond to glucose depletion. In contrast, phosphorylation and activation of Hsf1 upon heat shock was normal. The findings presented in this work thus provide a prime example of the multifunctionality of Hsp70 chaperones. Which of the functions of an Hsp70 leads to the observed phenotypic defects requires careful analysis.

## Methods

**Strains and plasmids.** Yeast strains and plasmids used in this study are listed in Supplementary Table 1. The parental strain MH272-3fa/α (ura3/ura3 leu2/leu2 his3/his3 trp1/trp1 ade2/ade2) and the haploid wild type, $\Delta ssb1\Delta ssb2$, $\Delta snf1$, and $\Delta ssb1\Delta ssb2\Delta snf1$ were as described[9]. $\Delta zuo1\Delta ssz1$ lacking RAC[54], and the strain expressing Ssb-K73A[32] were as described. $\Delta sse1$ was generated by replacing endogenous SSE1 with the HIS3 marker ± 300 bp. $\Delta sch9$ and $\Delta ssb1\Delta ssb2\Delta sch9$ were created by replacing the ApaI/XmaI fragment of endogenous SCH9 with the LEU2 marker ± 300 bp. The $\Delta sch9\Delta snf1$ strain was generated by mating $\Delta sch9$ with $\Delta snf1$ followed by tetrad dissection. The $\Delta yak1$ strain was constructed by replacing YAK1 with the yak1::kanMX cassette obtained from the Euroscarf deletion strain YI7006. The Hsf1-GFP strain was generated by replacing endogenous HSF1 with a C terminally GFP tagged version along with the klTRP1 auxotrophic marker. Sch9 low copy (pYCPlac33-Sch9, CEN, URA3) and high copy (pYEPlac195-Sch9, 2 µ, URA3) plasmids were generated by cloning SCH9 ± 300 bp into the respective vectors[55]. For phos-tag analysis, C terminally FLAG-tagged (DYKDDDDK) versions of Sch9 (Sch9-FLAG) were generated via PCR technology and were cloned into a low copy plasmid (pYCPlac33-Sch9-FLAG, CEN URA3) or high copy plasmid. Sch9-FLAG complemented growth defects of $\Delta sch9$ mutant strains (Supplementary Fig. 2b). Because the endogenous level of Sch9 is low (~ 400 molecules per cell[31]), co-immunoprecipitation and in vitro kinase experiments were performed in $\Delta sch9$ strains expressing Sch9-FLAG from a high copy plasmid (pYEPlac195-Sch9-FLAG, 2 µ, URA3) (Supplementary Fig. 3a). Sch9-2D3E-FLAG (T723D, S726D, T737E, S758E, S765E), Sch9-5A-FLAG (T723A, S726A, T737A, S758A, S765A), and Sch9-T570A-FLAG (T570A) were generated by replacing the MscI/XmaI fragments of pYCPlac33-Sch9-FLAG or pYEPlac195-Sch9-FLAG with the corresponding fragments from plasmids pJU841, pJU822, and pJU850[24]. Sch9-T570A/5A-FLAG was generated by replacing the Sac1/EcoR1 fragment of pYCPlac33-Sch9-5A-FLAG with the corresponding fragment from pYCPlac33-Sch9-T570A-FLAG. The Sac1/Xma1 fragments of Sch9-2D3E-FLAG and Sch9-5A-FLAG were used to replace the corresponding fragments of pYEPlac195-Sch9 to create non-tagged versions pYEPlac195-Sch9-2D3E and pYEPlac195-Sch9-5A. Sch9-K441A, Sch9-D556R, Sch9-T723A, Sch9-S726A, Sch9-T737A, Sch9-S758A, and Sch9-S765A were generated according to QuikChange (Agilent Technologies). The high copy plasmid for the overexpression of Ssa was generated by cloning SSA2 ± 300 bp into pYEPlac195. Yak1 was N-terminally FLAG-tagged via PCR technology and was cloned into a high copy plasmid (pYEPlac112-Yak1-FLAG, 2 µ, TRP1). An N-terminally His$_6$-sumo-tagged version of Hsf1 (pCA528-His$_6$-sumo-Hsf1) was employed for expression in E. coli[56]. The E. coli expression plasmid pGEX-4T-1-GST-Maf1 was previously described[41].

**Culture and harvest conditions.** Yeast strains were grown in YPD medium (1% yeast extract, 2% peptone, 2% glucose) or in SD minimal medium (6.7 g l$^{-1}$ yeast nitrogen base without amino acids, 2% glucose) supplemented with amino acids as required. Cultures were grown in liquid medium with constant shaking at 200 rpm at 30 °C. For glucose starvation experiments, logarithmically growing

YPD cultures were collected by centrifugation, cells were re-suspended in an equivalent volume of YP (1% yeast extract, 2% peptone), and were subsequently incubated at 30 °C with shaking at 200 rpm for 10 min, unless otherwise indicated. To starve cells for nitrogen, yeast strains grown in SD medium were collected by centrifugation, washed once with nitrogen starvation medium (SD-N: 6.7 g l$^{-1}$ yeast nitrogen base without amino acids and without ammonium sulphate, 2% glucose, supplemented with the required amino acids, buffered to pH ~ 6 with potassium phosphate buffer), re-suspended in an equal volume of SD-N and incubated at 30 °C with shaking at 200 rpm. To induce acute heat shock, logarithmically growing cells were collected by centrifugation, re-suspended in YPD preheated to 42 °C, and incubated at 42 °C for 15 min with shaking at 200 rpm. If indicated, cycloheximide (final concentration 25 µg ml$^{-1}$) or rapamycin (final concentration 500 ng ml$^{-1}$) was added to growing cultures 30 min prior to collection. Protein extracts were prepared according to[57]. In brief, 2–4 OD$_{600}$ units of cells were collected and re-suspended in 250–400 µl H$_2$0. Samples were mixed with an equal volume of 0.2 M NaOH and subsequently the suspension was incubated at room temperature for 5 min. NaOH-treated cells were collected by centrifugation, re-suspended in SDS sample buffer, boiled at 95 °C for 5 min, and cleared by centrifugation before loading.

**Phos-tag gel electrophoresis**. To avoid changes of protein phosphorylation patterns induced during collection, cells were heat-inactivated prior to collection, if total protein extracts were analyzed via phos-tag gels. Briefly, 3–4 ml of logarithmically growing cultures were directly transferred into test tubes immersed in a boiling water bath and were incubated for 4 min. Boiled cell cultures were subsequently cooled on ice, collected by centrifugation, and protein extracts[57] were dissolved in SDS sample buffer lacking EDTA.

Analysis of phosphorylation was performed employing phos-tag acrylamide (AAL-107, Wako Chemicals GmbH). SDS-PAGE and transfer to nitrocellulose membranes was performed according to the manufacturer's protocol, with minor modifications. Depending on the protein being resolved, the polyacrylamide concentration in the gel was adjusted to 4.5% for Sch9, 6% for Hsf1, and 8% for Maf1. Phos-tag gels contained 20 µM phos-tag reagent, or 30 µM in experiments which required a higher resolution of phosphorylated Sch9-5A-FLAG species. Gel electrophoresis was carried out at a constant current of 10 mA per gel (Bio-Rad Mini-PROTEAN, 1 mm). Manganese was removed by incubating gels in transfer buffer containing 10 mM EDTA for 7 to 10 min, followed by a washing step in transfer buffer without EDTA prior to blotting. Wet-transfer was performed at 375 mA for 2 h with cooling. As the available Sch9 antibody only poorly recognized endogenous, untagged phospho-forms of Sch9 on phos-tag gels, Sch9 and Sch9 mutants were FLAG-tagged and were expressed in a Δsch9 background for low copy plasmids for phos-tag analysis. Analysis of immunoblots derived from phos-tag gels was with α-FLAG (Sch9), α-Hsf1, or α-Maf1 as indicated. Molecular size markers are not indicated in Figures derived from phos-tag gels, as migration of proteins strongly depends on the phosphorylation level and must not correlate with the molecular mass. Supplementary Figs 7–12 show uncropped phos-tag gels with molecular size markers.

**Co-immunoprecipitation experiments**. ~Eighty OD$_{600}$ units of logarithmically growing YPD cultures of Δsch9 + pYEPlac195-Sch9 or Δsch9 + pYEPlac195-Sch9-FLAG were collected, cell pellets were re-suspended in lysis buffer I (20 mM HEPES-KOH, 150 mM potassium acetate, 2 mM magnesium acetate, 10% glycerol, 1× phosSTOP (Roche), 1 mM PMSF, 1 × PIC, pH 7.5), and were disrupted by the glass bead method as described[10]. Lysates were cleared by centrifugation at 14,000 r.p.m. for 5 min at 4 °C and lysate corresponding to 30–35 mg total protein (measured via A$_{280}$) was added to 25 µl α-FLAG-beads (anti-FLAG M2 Affinity Gel, Sigma-Aldrich) in a total volume of 1 ml. Binding to α-FLAG-beads was allowed for 1 h at 4 °C with gentle shaking. The beads were then washed for 5 min at 4 °C with 1.5 ml lysis buffer lacking phosSTOP and containing 0.1% Triton X-100, followed by a 1 ml washing step without the detergent. Washed α-FLAG-beads were boiled at 95 °C for 5 min in 40 µl SDS sample buffer to release bound proteins. Subsequent analysis was performed on 10% Tris-Tricine gels followed by immunoblotting employing α-Sch9, α-FLAG, α-Ssb, or α-Maf1 as indicated. For ATP-release experiments, co-immunoprecipitations were carried out as described above, except the second washing step was performed in the presence or absence of 10 mM ATP for 15 min at room temperature with gentle shaking. In order to estimate the extent of Sch9 overexpression in Δsch9 + pYEPlac195-Sch9-FLAG cells, glass bead lysate prepared from wild-type cells was compared with a dilution series of the Δsch9 + pYEPlac195-Sch9-FLAG lysate via immunoblotting using α-Sch9, followed by densitometric analysis. Protein concentrations in the extracts were adjusted according to the A$_{280}$ of the total lysates (Supplementary Fig. 3a). In order to estimate the amount of Sch9-FLAG bound to the FLAG-beads, Sch9 in cell lysates before and after binding to the FLAG-beads was compared.

To get a rough estimate of the fraction of Sch9-FLAG bound to Ssb or Maf1, we employed published copy-number estimates of yeast proteins (Sch9: ~400 molecules per cell; Maf1: 230 molecules/cell; Ssb 370.000 molecules per cell)[31]. Because Sch9-FLAG expression from pYEPlac195 was increased ~16-fold when compared to wild type (Supplementary Fig. 3a), the copy-number of Sch9-FLAG in the Δsch9 + pYEPlac195-Sch9-FLAG strain was ~ 6400 molecules per cell. Roughly 75% of total Sch9-FLAG was recovered bound to the α-FLAG-beads

(Supplementary Fig. 3b), this amounts to ~ 4800 Sch9-FLAG molecules per cell recovered on α-FLAG-beads. About 0.05% of total Ssb (~190 molecules per cell) and 0.05% of total Maf1 (0.12 molecules per cell) were co-immunoprecipitated together with Sch9-FLAG. From these estimates we calculated that roughly 4% of Sch9-FLAG was associated with Ssb, and 0.0025% of Sch9-FLAG was associated with Maf1 in the experiment shown in Fig. 3.

**Aggregation assay**. Insoluble proteins were isolated as described[12] with minor modifications. Briefly, cell cultures corresponding to 25–50 OD$_{600}$ units were collected and re-suspended in lysis buffer II (20 mM sodium phosphate pH 6.8, 10 mM DTT, 1 mM EDTA, 0.1% Tween-20 1.25 U ml$^{-1}$ benzonase, 1 mM PMSF, 1 x PIC) in a volume of 1.5 ml, 0.1 mg ml$^{-1}$ zymolyase T20 per OD$_{600}$-unit was added, and the suspension was incubated for 30 min at 30 °C. Cell suspensions were subsequently sonicated (Vibra-Cell; Amplitude 40%, Pulse on/off: 0.5 s/0.5 s, for 10 s) followed by a clearing spin at 200×g for 20 min at 4 °C. Equal amounts of supernatants (termed total lysates) according to A$_{280}$ in a total volume of 1 ml were centrifuged at 16.000 g for 20 min at 4 °C. Next 1.5 ml of 20 mM sodium phosphate pH 6.8 buffer containing 1 mM PMSF, 1× PIC, 2% NP-40 (NP-40 Alternative, Calbiochem) was added to the resulting pellets, which were subsequently re-suspended by sonication (amplitude 40%, pulse on/off: 0.5 s/0.5 s, for 6 s) and centrifugation at 16,000×g for 20 min at 4 °C. The procedure was repeated in the same buffer lacking NP-40 followed by sonication (amplitude 22%, pulse on/off: 0.6 s/0.4 s, for 4 s) and centrifugation at 16,000×g for 20 min at 4 °C. After the second wash, pellets (termed insoluble material or aggregates) and aliquots of the total lysates were precipitated with 5% trichloroacetic acid (TCA) and were subsequently dissolved in sample buffer. Analysis was performed via 10% Tris-Tricine gels followed by Coomassie staining or immunoblotting.

**Partial purification of GST-Maf1 and His$_6$-sumo-Hsf1**. Expression of GST-Maf1 was as described[41] with minor changes. E. coli BL21 cells transformed with the plasmid pGEX-4T-1-GST-Maf1 (GST-Maf1) were induced with 0.1 mM IPTG for 3 h at 37 °C. ~100 OD$_{600}$ unit of cells were collected, re-suspended in 1.5 ml lysis buffer III (25 mM Tris-HCl, 150 mM NaCl, 1 mM EDTA, 1 mM PMSF, 1× PIC, pH 7.5) and were lysed by sonication (Vibra-cell; Amplitude: 35%, Pulse on/off: 1 s/1 s, for 30 s × 4 cycles). Lysate was cleared by centrifugation at 14,000 r.p.m. for 20 min at 4 °C, and then incubated with 50 µl glutathione sepharose beads (Pharmacia Biotech) for 1 h at 4 °C with gentle shaking. Beads were washed twice with 1 ml lysis buffer III containing 0.5% Triton X-100 for 5 min at 4 °C. Elution was performed by incubating the beads with 500 µl elution buffer I (50 mM Tris/HCl, 5% glycerol, 1× PIC, pH 8.0) containing 10 mM reduced glutathione for 15–20 min at room temperature. E. coli BL21 transformed with pCA528-His$_6$-sumo-Hsf1 (His$_6$-sumo: MGHHHHHHGSDSEVNQEAKPEVKPEVK-PETHINLKVSDGSSEIFFKIKKTTPLRRLMEAFAKRQGKEMDSLR-FLYDGIRIQADQTPEDLDMEDNDIIEAHREQIGG) was induced with 1 mM IPTG for ~4 h at 20 °C. Cells corresponding to ~ 50–100 OD$_{600}$ units were collected and lysed by sonication as described above in 1.5 ml lysis buffer IV (40 mM HEPES-KOH, 240 mM potassium acetate, 5 mM magnesium acetate, 15 mM imidazole, 1 mM PMSF, 1× PIC, pH 7.8). Lysate was cleared by centrifugation at 14,000 r.p.m. for 20 min at 4 °C, and were then incubated with 50 µl Ni-NTA agarose beads (Qiagen) for 1.5 h at 4 °C with gentle shaking. Ni-beads were washed twice with 1 ml lysis buffer IV complemented with 480 mM potassium acetate and 30 mM imidazole for 10 min at 4 °C. Elution was performed by incubating the Ni-beads with lysis buffer IV containing 200 mM imidazole for 20 min at room temperature with gentle shaking. The buffer of the His$_6$-sumo-Hsf1 eluate was subsequently exchanged for elution buffer I using PD-10 G25 sephadex columns (GE Healthcare Life Sciences). The eluate was finally concentrated ~10-fold using micro spin columns (Microcon, YM-30). Partly purified proteins are shown in Supplementary Fig. 5.

**In vitro kinase assays**. ~ 40 OD$_{600}$ units of logarithmically growing Δsch9 + pYEP195-Sch9-2D3E-FLAG or Δsch9 + pYEP195-Sch9-5A-FLAG cells were collected, disrupted by the glass bead method in lysis buffer I, cleared, and subsequently incubated with α-FLAG-beads as described above (Supplementary Fig. 5). Kinase reactions were then carried out with Sch9-2D3E-FLAG or Sch9-5A-FLAG bound to α-FLAG-beads. To that end, beads were incubated with a mixture of ~0.5 µg partially purified His$_6$-sumo-Hsf1 and ~ 0.5 µg GST-Maf1 (Supplementary Fig. 5) in a total volume of 50 µl kinase buffer containing 50 µCi γ-$^{32}$P-ATP in kinase buffer (1× PBS, 100 µM ATP, 20% glycerol, 4 mM magnesium chloride, 10 mM DTT, 1× PIC, pH 7.5). Reactions were incubated for 15 min at 30 °C, supernatants containing His$_6$-sumo-Hsf1 and GST-Maf1 were collected, precipitated with TCA, separated on 10% Tris-Tricine gels, stained with Coomassie, and subsequently exposed to a phosphor image plate.

**RT-PCR**. Primers used for the RT–PCR reactions are listed in Supplementary Table 2. ~15 OD$_{600}$ units of logarithmically growing YPD cultures were collected and total RNA was extracted with the RNeasy Mini Kit (Qiagen). One µg of RNA was reverse-transcribed to cDNA using the iScript cDNA synthesis kit (Bio-Rad). The cDNA was then used as a template to amplify 120–150 bp-long fragments

within the open reading frames of the target genes. PCR products were analyzed on agarose gels, using a 630 bp fragment of *ACT1* as a loading control.

**Antibodies**. α-Hsf1 (1:5000; Supplementary Fig. 4a and Supplementary Note 3), α-Sch9 (1:500; Supplementary Fig. 6a), α-Sse1 (1:10,000)[9], α-Ssb (1:5000)[58], α-Rpl35 (1:10,000)[4], α-Rpl17 (1:10,000)[58], α-Rpl19 (1:10,000)[4], α-Rps6 (1:10,000; Supplementary Fig. 6b), α-Rps9 (1:10,000)[58], α-Ssa (1:10,000)[9], α-Pgk1 (1:5000; Supplementary Fig. 6c), and α-Rps20 (1:10,000; Supplementary Fig. 6d) were rabbit polyclonal antibodies (Eurogentec, Rospert lab antibody collection). The rabbit polyclonal α-Maf1 (1:5000) was a kind gift from Olivier Lefebvre (Commissariat à l'Energie Atomique, France). α-phospho-Sch9-T570 (1:5000)[24] was kindly provided by Robbie Loewith (University of Geneva). Commercially purchased antibodies were α-FLAG (1:10,000; Agilent 200471), α-GFP (1:5000; Life Technologies A11122), α-rabbit HRP (1:10,000; Pierce 31460) and α-mouse HRP (1:10,000; Santa Cruz SC-2005). Immunoblots were detected by enhanced chemiluminescence[59]. Uncropped scans of the most important immunoblots are shown in Supplementary Figs 7–12.

**Miscellaneous**. Protease inhibitor cocktail (1000× PIC) contained 1.25 mg ml$^{-1}$ leupeptin, 0.75 mg ml$^{-1}$ antipain, 0.25 mg ml$^{-1}$ chymostatin, 0.25 mg ml$^{-1}$ elastinal, and 5 mg ml$^{-1}$ pepstatin-A in DMSO. Lambda phosphatase (New England Biolabs) and alkaline phosphatase (FastAP, Thermo Fischer Scientific) treatment was performed in total cell lysates derived from 10–20 OD$_{600}$ units of cells disrupted by the glass bead method in lysis buffer I (without phosSTOP and glycerol) according to the manufacturer's protocol. Subsequently, proteins were precipitated with TCA and were dissolved in SDS sample buffer.

**Data availability**. All relevant data are available from the authors upon request.

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

## Acknowledgements

This work was supported by Deutsche Forschungsgemeinschaft [SFB 746] to S.R., [DFG RO 1028/5-1] to S.R, by the Excellence Initiative of the German federal and state governments [BIOSS-2] to S.R., and by [SFB 1036 TP9] to M.P.M. The rabbit polyclonal α-Maf1 was a kind gift from Olivier Lefebvre (Commissariat à l'Energie Atomique, France). α-phospho-Sch9-T570 and plasmids pJU841, pJU822, and pJU850 were kindly provided by Robbie Loewith (University of Geneva). The *E. coli* expression plasmid pGEX-4T-1-GST-Maf1 was a kind gift from X.F. Steven Zheng (Rutgers-CINJ).

## Author contributions

K.M.: methodology, investigation, validation, formal analysis, visualization, writing of original draft. M.P.M.: methodology, writing review and editing. E.F.: investigation, validation. C.P.: investigation, validation. S.R.: conceptualization, validation, visualization, writing of original draft, supervision, project administration, funding acquisition.

## Additional information

**Competing interests:** The authors declare no competing financial interests.

