## [Peer Review File · Nature Communications]

Reviewers' Comments:

Reviewer #1 (Remarks to the Author):

In addition to the ubiquitous cytosolic Hsp70 chaperones (Ssa1-4), yeast possess a dedicated pair of Hsp70 proteins that associate with the ribosome to facilitate de novo protein folding. Recent work also demonstrated that Ssb chaperones are required for proper ribosome biogenesis, including assembly and nuclear export by an unknown mechanism. Ribosome biogenesis is also regulated via the TORC1-Sch9 kinase cascade, in part through Sch9-dependent regulation of the Pol III transcriptional repressor Maf1. In this report, Mudholkar et al reveal that these two pathways intersect, demonstrating that Ssb interacts with Sch9 and is required for Sch9 phosphorylation by TORC1. Consistent with this discovery, cells lacking either Sch9 or both Ssb proteins exhibit similar phenotypes with respect to ribosome biogenesis, including aggregation of ribosomal subunits. Additionally, Sch9 is found to promote Hsf1 activity via direct phosphorylation in an Ssb-dependent manner, suggesting that Sch9 may rely on chaperone modulation for multiple cellular activities.

This study relies heavily on the Phos-Tag technology to minutely dissect differential phosphorylation of Sch9 to a level not previously achieved. As such, the authors could discern differences in phosphorylation that permit the convincing demonstration that Ssb is required for both phosphorylation and subsequent kinase activity of Sch9. The immunoprecipitation-based demonstration that Ssb binds Sch9 in an ATP-dependent manner is somewhat less convincing but when considered in context with the rest of the story is reasonable. With the exception of some unanswered questions detailed below, this work expands our understanding of how chaperones promote ribosomal biogenesis and unites what were previously thought to be independent pathways.

1. In terms of mechanism, the authors speculate that Ssb may be facilitating access to one or more TORC1 phosphorylation sites in the so-called turn domain of Sch9. Why would Ssb be playing this role, and not the Ssa proteins? Have the authors asked whether Ssa also co-precipitates with Sch9? Can overexpressed Ssa or possibly the well-established Craig lab Ssa-Ssb hybrids substitute for Ssb in this role? If not, why?

2. Given the detail of phosphorylation analysis done with Sch9 in the paper, it was surprising to

not see the Ssb-dependent residues identified via mutagenesis or mass spectrometry, especially since the abstract explicitly states "...specific residues within the C-terminus...". Conclusively identifying the residues as those in the turn domain would bolster the model proposed in the discussion and seems relatively straightforward given the work already done.

3. Is the requirement for Ssb to promote Sch9 phosphorylation absolute? Can overexpression of SCH9 suppress any of the defects of an *ssb1Δssb2Δ* mutant?

4. In the context of the now possible role for Sch9 and Ssb in regulating Hsf1 activity, the authors should review the 2000 paper from Bonner and colleagues that identifies Ssb as a key modulator of Hsf1 function – these results have been hard to explain for years and the idea that Ssb and Sch9 might exist in a stable physical complex with Hsf1 in unstressed cells to prime it for activation is intriguing.

p. 5, top paragraph – Fig 4c incorrectly referenced as 4d

Reviewer #2 (Remarks to the Author):

Mudholkar et al demonstrate that the chaperone Ssb functions in ribosome biogenesis by acting as a scaffold that allows the TORC1 target Sch9 to be phosphorylated and activated. The function of TORC1/Sch9 in ribosome biogenesis has been previously defined. In the current studies, they then add Ssb as an important component to this TORC1/Sch9 function. Furthermore, previous studies have elucidated a role for Ssb in cotranslational folding but whether it has a role in ribosome assembly has remained unclear. Findings from this current study provide evidence to support a role for the latter.

First, they analyzed phosphorylation of Sch9 under glucose-replete or -deplete conditions in the presence or absence of *ssb*. They found that Sch9 becomes slightly hypophosphorylated in the absence of *ssb* when glucose is present. Upon glucose starvation, a hyperphosphorylated form of Sch9 that is attributed to Snf1, is absent in the *ssb* mutant. A small fraction of Ssb was also found to interact with Sch9 in the absence of ATP. These findings suggested that the fraction of Sch9 that undergoes hyperphosphorylation during glucose starvation requires Ssb.

Second, they analyzed downstream targets of Sch9 and found that an *ssb* deletion mutant shares similar phenotype with *sch9* mutant, such as decreased Maf1 phosphorylation, increased aggregation of ribosomal proteins.

Third, they examined how Hsf1 phosphorylation and activation could be regulated by Ssb and

Sch9. They found that its hyperphosphorylation during glucose withdrawal is also dependent on Sch9 and that Sch9 can directly phosphorylate it. The expression of Hsf1-controlled genes was defective upon glucose starvation in cells lacking *ssb* or *sch9*, thus correlating with the defective Hsf1 phosphorylation.

Overall, the data are mostly supportive of the model they are proposing. Specific comments below should be addressed to strengthen the conclusions of the study.

1. The fraction of hyperphosphorylated species of Sch9 that are dependent on Ssb1 (or Snf1 in Fig 1) seems to be quite small. Glucose withdrawal was conducted only from 10-20 min. Would prolonged starvation have a more striking effect on the phosphorylation? Would starvation with nitrogen have a more pronounced effect on Ssb-mediated phosphorylation?

If not, they should quantitate the fraction that gets hyperphosphorylated and include in the discussion why they think only a small fraction would be dependent on Ssb.

2. The statement that "Ssb interacts with Sch9 in an ATP-dependent manner" should be clarified. It is perhaps better to state "...in an ATP-sensitive manner" since their interaction is abolished by ATP.

3. There is only indirect data to support that Ssb affects ribosome biogenesis. It seems that the story revolves around the issue that Ssb is regulating Sch9 and thus downstream functions of Sch9 including possibly ribosome biogenesis and Hsf1 to some extent require Ssb. The authors should either add more supporting data on the effect of Ssb on ribosome biogenesis or modify the title of the study.

4. In Fig 5, again only a fraction of Hsf1 undergoes hyperphosphorylation in a *ssb*-dependent manner. To support that the H3 phosphorylation in *ssb* mutants is due to Sch9, they should express Sch9-5A mutant in the *ssb*-delete and *sch*-delete background.

5. The slow-migrating species of Hsf under heat-shock conditions should be verified using phosphatase treatment.

Reviewer #3 (Remarks to the Author):

This manuscript reports a possible rethinking of the function of the chaperone Ssb in ribosome biogenesis. Whereas it Ssb been regarded to work largely on misfolded proteins or to properly fold proteins, here, the authors suggest that Ssb promotes TORC1 phosphorylation of Sch9.

While this may be the case, the work is largely descriptive and reports correlations. Convincing functional evidence is needed to support the authors' conclusions before the work is acceptable for publication. In addition, there are some issues with presentation.

1. The authors suggest that Ssb provides a scaffold function for TORC1 phosphorylation of Sch9. This is the crux of the paper but is not well substantiated. Does Ssb modulate TORC1 phosphorylation of Sch9 *in vitro*? Does it require the formation of a heterotrimeric complex? Can the C-terminus of Sch9 be moved to a different carrier protein such as GST and now be phosphorylated without the need for Ssb? Additional experiments along these should be included.
2. The authors show that Ssb interacts with Sch9 in an ATP-dependent manner, suggesting that interaction requires its chaperone function, albeit not co-translationally. This seems counter to their suggestion that it serves as a scaffold, which implies a simple structural role.
3. If Ssb is needed for TORC1 phosphorylation of Sch9, can the *ssb* Δ effect be bypassed by a Sch9 phosphomimic? Can a phosphomimic of Maf1 bypass the defect that *ssb* Δ causes with Sch9?
4. The authors are missing opportunities to test their models by genetic interactions. For example, they argue that aggregation of ribosomal proteins in Δ *ssb* cells is due to loss of activity of Sch9. If this were true, *ssb* and *sch9* should be epistatic. Similarly, a phosphomimic of Sch9 should bypass the ribosome protein aggregation defect seen in *ssb* Δ cells. Such experimental approaches should be included.
5. L123. The authors should explain how they made the connection between Ssb and TORC1 phosphorylation of Sch9.
6. L363 The authors report that Δ *ssb* does not induce a misfolded protein response because Hsf1 is not phosphorylated. However, they also claim that Ssb is needed for Sch9 signaling which is needed for Hsf1 phosphorylation. So, activation of Hsf1 would not be expected and cannot be used as a proxy for a misfolded protein response. The logic here is circular.

Reply to reviewers' comments

Reviewer #1 (Remarks to the Author):

In addition to the ubiquitous cytosolic Hsp70 chaperones (Ssa1-4), yeast possess a dedicated pair of Hsp70 proteins that associate with the ribosome to facilitate de novo protein folding. Recent work also demonstrated that Ssb chaperones are required for proper ribosome biogenesis, including assembly and nuclear export by an unknown mechanism. Ribosome biogenesis is also regulated via the TORC1-Sch9 kinase cascade, in part through Sch9-dependent regulation of the Pol III transcriptional repressor Maf1. In this report, Mudholkar et al reveal that these two pathways intersect, demonstrating that Ssb interacts with Sch9 and is required for Sch9 phosphorylation by TORC1. Consistent with this discovery, cells lacking either Sch9 or both Ssb proteins exhibit similar phenotypes with respect to ribosome biogenesis, including aggregation of ribosomal subunits. Additionally, Sch9 is found to promote Hsf1 activity via direct phosphorylation in an Ssb-dependent manner, suggesting that Sch9 may rely on chaperone modulation for multiple cellular activities.

This study relies heavily on the Phos-Tag technology to minutely dissect differential phosphorylation of Sch9 to a level not previously achieved. As such, the authors could discern differences in phosphorylation that permit the convincing demonstration that Ssb is required for both phosphorylation and subsequent kinase activity of Sch9. The immunoprecipitation-based demonstration that Ssb binds Sch9 in an ATP-dependent manner is somewhat less convincing but when considered in context with the rest of the story is reasonable. With the exception of some unanswered questions detailed below, this work expands our understanding of how chaperones promote ribosomal biogenesis and unites what were previously thought to be independent pathways.

We thank the referee for this positive evaluation of our work. We also thank for suggesting excellent additional experiments, which we performed. We feel the new data strengthen our conclusions and provide additional insight into the role of Ssb in TORC1-Sch9 signaling.

1. In terms of mechanism, the authors speculate that Ssb may be facilitating access to one or more TORC1 phosphorylation sites in the so-called turn domain of Sch9. Why would Ssb be playing this role, and not the Ssa proteins? Have the authors asked whether Ssa also co-precipitates with Sch9? Can overexpressed Ssa or possibly the well-established Craig lab Ssa-Ssb hybrids substitute for Ssb in this role? If not, why?

Yes, this is an excellent point. We tested the effect of Ssa with respect to the regulation of Sch9 activity. The data are now shown in Fig. 3f, 3h and Fig. S3e. In a nutshell: Ssa also associates with Sch9 as judged from co-IP experiments (Fig. S3e). Because Ssa is essential we were unable to analyze Sch9 activity in the absence of Ssa. However, we have now shown that overexpression of Ssa does not rescue Sch9-activity in the $\Delta ssa1\Delta ssa2$ strain (Fig. 3h). We also show that Sch9 activity is compromised in the absence of the Ssb-cochaperone RAC, while Sse1, which *in vivo* predominantly participates in Ssa-dependent processes, is *not* required for Sch9 activity (Fig. 3f). We conclude that Ssb affects Sch9 function, while Ssa does not. We feel that this is not too surprising. The C-terminal domain of Ssb is not conserved in other Hsp70 homologs including Ssa. Of note, Ssb does not only interact with ribosomes via its very C-terminal helix^{1,2}, but also with the 14-3-3 homolog Bmh1³. Our current model is that the very C-terminus of Hsp70s connects the otherwise homologous chaperones to distinct networks of partner chaperones, which then provide functional specificity (see also⁴).

2. Given the detail of phosphorylation analysis done with Sch9 in the paper, it was surprising to not see the Ssb-dependent residues identified via mutagenesis or mass spectrometry,

especially since the abstract explicitly states "...specific residues within the C-terminus...". Conclusively identifying the residues as those in the turn domain would bolster the model proposed in the discussion and seems relatively straightforward given the work already done. We now identified residue T737 within the C-terminal domain of Sch9 via mutational analysis as the residue, which requires Ssb for efficient TORC1-dependent phosphorylation. The data are shown in Fig. 2d and Fig. 3f. Of note, the single T737A mutant is the only mutation of the 5 TORC1-dependent residues, which confers a growth defect (Fig. S2a)⁵.

3. Is the requirement for Ssb to promote Sch9 phosphorylation absolute? Can overexpression of SCH9 suppress any of the defects of an *ssb1Δssb2Δ* mutant?

We have tested if SCH9 overexpression rescues Maf1 phosphorylation in the *Δssb1Δssb2* strain. The answer is: yes it does. The data are now included in Fig. 3d (right panel). Based on the data we conclude that Ssb is not absolute, but can be overcome by overexpression of Sch9.

4. In the context of the now possible role for Sch9 and Ssb in regulating Hsf1 activity, the authors should review the 2000 paper from Bonner and colleagues that identifies Ssb as a key modulator of Hsf1 function – these results have been hard to explain for years and the idea that Ssb and Sch9 might exist in a stable physical complex with Hsf1 in unstressed cells to prime it for activation is intriguing.

We now reference and discuss the work of Bonner et al. ⁶.

p. 5, top paragraph – Fig 4c incorrectly referenced as 4d

done

Reviewer #2 (Remarks to the Author):

Mudholkar et al demonstrate that the chaperone Ssb functions in ribosome biogenesis by acting as a scaffold that allows the TORC1 target Sch9 to be phosphorylated and activated. The function of TORC1/Sch9 in ribosome biogenesis has been previously defined. In the current studies, they then add Ssb as an important component to this TORC1/Sch9 function. Furthermore, previous studies have elucidated a role for Ssb in cotranslational folding but whether it has a role in ribosome assembly has remained unclear. Findings from this current study provide evidence to support a role for the latter.

First, they analyzed phosphorylation of Sch9 under glucose-replete or -deplete conditions in the presence or absence of *ssb*. They found that Sch9 becomes slightly hypophosphorylated in the absence of *ssb* when glucose is present. Upon glucose starvation, a hyperphosphorylated form of Sch9 that is attributed to Snf1, is absent in the *ssb* mutant. A small fraction of Ssb was also found to interact with Sch9 in the absence of ATP. These findings suggested that the fraction of Sch9 that undergoes hyperphosphorylation during glucose starvation requires Ssb.

We think that this part of the data has been slightly misinterpreted, and we realize that our writing may not have been clear. We have tried to make that part more clear now in the Results section by adding the missing basic information as well as some new data. Specifically Sch9 phosphorylation after glucose depletion is now described in more detail. We are sorry for not having been clear in the initial version of the manuscript. In a nutshell:

In the presence of glucose

Our data reveal that in wild type cells in the presence of glucose the bulk of Sch9 is phosphorylated at all of the five TORC1-dependent residues, which then migrate as S3 (Fig.

1b). This is consistent with previous findings, which have employed different methods to determine Sch9 phosphorylation⁵. We now show that the change of Sch9 phosphorylation in cells lacking Ssb is due to the lack of one single TORC1-dependent phosphorylation, which occurs on residue T737 and results in a quantitative down-shift of S3 to S3[#] (Fig. 2d). This "slight hypo-phosphorylation" (or in other words, this small shift on the phos-tag gel), therefore, applies to the bulk of Sch9 molecules in cells lacking Ssb. As phosphorylation of Sch9-T737 is required for the activation of Sch9, the lack of this single phosphorylation negatively affects Sch9 function (new Figs. 3f and S2a, see also⁵. See also comment to referee 1).

In the absence of glucose

We are sorry for being unclear with respect to this point. Sch9 phosphorylation upon glucose depletion is now described in more detail in the Results section. We do not want to suggest that "the fraction of Sch9 that undergoes hyperphosphorylation during glucose starvation requires Ssb". Our data indicate that Snf1-dependent phosphorylation of Sch9, which occurs in the absence of glucose, is *not* affected by Ssb (Fig. 2e). The reason that S2 and S4 migrate faster as S2[#] and S4[#] in $\Delta ssb1\Delta ssb2$ (Fig. 2f) is due to the fact that phosphorylation on Sch9-T737 is constitutively absent in $\Delta ssb1\Delta ssb2$ both in the presence (S3[#]) and absence (S2[#], S4[#]) of glucose. However, SNF1-dependent phosphorylation occurs in both wild type and $\Delta ssb1\Delta ssb2$ (Fig. 2e). Please also note, it is not a small fraction of Sch9, which is phosphorylated in an SNF1-dependent manner upon glucose starvation, but the bulk of Sch9 becomes phosphorylated in an SNF1-dependent manner. All three Sch9 forms, S0, S1 and S3 are phosphorylated upon glucose starvation in an SNF1-dependent manner (Fig. 1e, Fig. S1a). This is evident from the SNF1-dependent phosphorylation of the Sch9-T570/5A mutant upon glucose depletion shown in the new Fig. 1e.

Second, they analyzed downstream targets of Sch9 and found that an *ssb* deletion mutant shares similar phenotype with *sch9* mutant, such as decreased Maf1 phosphorylation, increased aggregation of ribosomal proteins.

Third, they examined how Hsf1 phosphorylation and activation could be regulated by Ssb and Sch9. They found that its hyperphosphorylation during glucose withdrawal is also dependent on Sch9 and that Sch9 can directly phosphorylate it. The expression of Hsf1-controlled genes was defective upon glucose starvation in cells lacking *ssb* or *sch9*, thus correlating with the defective Hsf1 phosphorylation.

Overall, the data are mostly supportive of the model they are proposing. Specific comments below should be addressed to strengthen the conclusions of the study.

1. The fraction of hyperphosphorylated species of Sch9 that are dependent on Ssb1 (or Snf1 in Fig 1) seems to be quite small.

As explained above and now also more precisely described in the text our data indicate that the bulk of Sch9 molecules lacks phosphorylation at T737 in glucose-grown cells lacking Ssb. In glucose-depleted cells the bulk of Sch9 molecules lacks the SNF1-dependent phosphorylation when Snf1 is absent, whereas Ssb does not affect the Snf1-dependent phosphorylation of Sch9.

Glucose withdrawal was conducted only from 10-20 min. Would prolonged starvation have a more striking effect on the phosphorylation? Would starvation with nitrogen have a more pronounced effect on Ssb-mediated phosphorylation?

Fig. R1

The effect on phosphorylation upon glucose withdrawal is quite striking, even after 15 min (Fig. 2f) and concerns the bulk of Sch9 molecules. The above Fig. R1 shows a 1h glucose starvation time course. It seems that the TORC1-dependent phosphorylation of Sch9 is partly restored after longer starvation times (reappearance of S4 and S4# after 60 min). These data are interesting, however, do not impact the conclusions of this study. Moreover, Ssb does not affect the reappearance of the TORC1-dependent phosphorylation after prolonged starvation. In the absence of Ssb the bulk of Sch9 species (S2# and S4#) migrate more quickly than in the wild type because Sch9-T737 is non-phosphorylated at any of the time points during the time course.

We now also performed nitrogen starvation experiments and found (as expected) that upon nitrogen starvation Sch9 is dephosphorylated at all 5 TORC1-dependent sites. The dephosphorylation is unaffected by Ssb (in case Ssb is absent, the remaining 4 TORC1-dependent phosphorylations are removed from Sch9). In contrast to glucose depletion, however, Sch9 is not, at the same time, phosphorylated at other residues. This is consistent with the idea that SNF1 is not activated under these conditions. The new data are shown in Fig. S1b.

If not, they should quantitate the fraction that gets hyperphosphorylated and include in the discussion why they think only a small fraction would be dependent on Ssb.

See above.

Most of the Sch9 molecules lack phosphorylation at Sch9-T737 in the absence of Ssb, and most of Sch9 is hyper-phosphorylated in an SNF1-dependent manner.

2. The statement that "Ssb interacts with Sch9 in an ATP-dependent manner" should be clarified. It is perhaps better to state "...in an ATP-sensitive manner" since their interaction is abolished by ATP.

Thank you for pointing this out correctly. We have changed ATP-dependent to ATP-sensitive.

3. There is only indirect data to support that Ssb affects ribosome biogenesis. It seems that the story revolves around the issue that Ssb is regulating Sch9 and thus downstream functions of Sch9 including possibly ribosome biogenesis and Hsf1 to some extent require Ssb. The authors should either add more supporting data on the effect of Ssb on ribosome biogenesis or modify the title of the study.

We would like to point out that the role of Ssb in ribosome assembly is ill-defined. However, that Ssb plays such a role was - as we feel, convincingly - shown by previous work from other labs^{7,8}. Our work now reveals that one, but not the only, function of Ssb with respect to ribosome biogenesis is connected to the regulation of the signaling kinase Sch9.

We made that point now more clear in the Introduction.

4. In Fig 5, again only a fraction of Hsf1 undergoes hyperphosphorylation in a *ssb*-dependent manner. To support that the H3 phosphorylation in *ssb* mutants is due to Sch9, they should express Sch9-5A mutant in the *ssb*-delete and *sch*-delete background.

In the presence of glucose

The phos-tag data reveal that the bulk of Hsf1 remains hypo-phosphorylated when Ssb is absent. The H1* species is less phosphorylated compared to H1, the H2* species is less phosphorylated compared to H2 (Fig. 5a, Fig. 6a and Fig. 6b; Fig. S4c). We have tried to make that more clear in the text. We also expressed Sch9-5A in the $\Delta sch9$ mutant. The data reveal that the bulk of Hsf1 remains hypo-phosphorylated and the phosphorylation pattern of Hsf1 is similar when Sch9 is absent or when cells express Sch9-5A (Fig. 5a, S4e). We have now also expressed Sch9-5A and in addition the constitutively active Sch9-2D3E mutant in a $\Delta ssb1\Delta ssb2$ background (Fig. S4d). Sch9-5A does not rescue Hsf1 hypo-phosphorylation in the $\Delta ssb1\Delta ssb2$ strain; however, Sch9-2D3E does (Fig. S4d). These data strongly support the idea that Hsf1 is hypo-phosphorylated in the absence of Ssb, because Sch9 is not properly activated.

In the absence of glucose

The bulk of the Hsf1 H1/H2 species (which are both phosphorylated at unknown sites (Fig. S4b) become phosphorylated in a Yak1-dependent manner (Fig. 5c). This results in the formation of species migrating as a poorly resolved double-band, we collectively termed H3 (e.g. Fig. S4c). The phos-tag analysis strongly suggests that in the wild type upon glucose depletion the H1 and H2 species are almost quantitatively phosphorylated, resulting in the H3 species. H3 cannot form in the absence of Sch9 or Ssb since the H1*/H2* species (which are hypo-phosphorylated due to lack of Sch9 activity; see above) are not a good substrate for Yak1-dependent phosphorylation.

Below we show (Fig. R2) an extended version of Fig. S4d for the information of the referee. However, we feel that the data require additional information, which we would like to omit from the paper, because the result does not impact on the conclusions of the manuscript.

Fig. R2. In the presence of glucose, overexpression of Sch9-2D3E in $\Delta ssb1\Delta ssb2$ rescued the phosphorylation defect of Hsf1, i.e. H1*/H2* is again phosphorylated to H1/H2, like in the wild type. This is not the case if the Sch9-5A mutant is expressed in the $\Delta ssb1\Delta ssb2$ background. Upon glucose depletion, however, expression of Sch9-2D3E did not lead to Hsf1-H3 formation in the $\Delta ssb1\Delta ssb2$. This result is at first sight unexpected, however, is explained by previously published observations. Yak1, which is responsible for the phosphorylation of Hsf1-H1/H2 to Hsf1-H3, is negatively regulated by Sch9. First, inhibition of the TORC1/Sch9 pathway by treatment with rapamycin leads to activation and nuclear accumulation of Yak1⁹. Second, in $\Delta sch9$ cells and in cells expressing the inactive Sch9-5A Yak1 is maintained in an active state¹⁰. Therefore, the Sch9-2D3E, which cannot be inactivated and remains active during glucose depletion, is expected to constitutively repress

Yak1 activity and thus prevent Hsf1-H3 formation. These findings do not impact on the role of Ssb in the activation of Sch9.

5. The slow-migrating species of Hsf under heat-shock conditions should be verified using phosphatase treatment.

The experiment was done and is now shown in Fig. S4b.

Reviewer #3 (Remarks to the Author):

This manuscript reports a possible rethinking of the function of the chaperone Ssb in ribosome biogenesis. Whereas it Ssb been regarded to work largely on misfolded proteins or to properly fold proteins, here, the authors suggest that Ssb promotes TORC1 phosphorylation of Sch9. While this may be the case, the work is largely descriptive and reports correlations. Convincing functional evidence is needed to support the authors' conclusions before the work is acceptable for publication. In addition, there are some issues with presentation.

1. The authors suggest that Ssb provides a scaffold function for TORC1 phosphorylation of Sch9. This is the crux of the paper but is not well substantiated. Does Ssb modulate TORC1 phosphorylation of Sch9 *in vitro*? Does it require the formation of a heterotrimeric complex? Can the C-terminus of Sch9 be moved to a different carrier protein such as GST and now be phosphorylated without the need for Ssb? Additional experiments along these should be included.

We agree with the referee: it would be very interesting to study the effect of Ssb on TORC1-dependent Sch9 phosphorylation *in vitro* using purified components. However, such experiments are far from being straightforward and would go beyond the scope of this study. One difficulty with setting up an *in vitro* system is that as yet unidentified components might be involved in function of Ssb in Sch9 regulation. Our own unpublished data suggest that this might be the case. Of note, the role of Ssb in the regulation of the kinase SNF1 is connected to the 14-3-3 protein Bmh1, which interacts with Ssb^{3,4}. Moreover, we have preliminary evidence that the role of Ssb in signaling *versus* translation is regulated via phosphorylation of Ssb, which further complicates the design of an *in vitro* experimental system. Moreover, we now show in the new Fig. 3d (right panel) that *in vivo* the requirement of Ssb is not absolute, and that overexpression of Sch9 rescues the Maf1 phosphorylation defect of $\Delta ssb1\Delta ssb2$ cells. As the concentrations *in vitro* differ from those in a cell this might lead to a situation in which Sch9 phosphorylation becomes independent of Ssb.

We have now performed additional experiments to determine if the function of Ssb resembles a static scaffolding protein. However, this is not the case. We now show that the activation of Ssb requires the Ssb J-domain partner RAC and requires the ATPase activity of Ssb (Fig. 3f and 3g). This is also consistent with the finding that Ssb interacts with Sch9 in an ATP-sensitive manner and suggests that Sch9 cycles on and off Ssb. We make that point more clearly in the text. Moreover, we now discuss a recent paper, which revealed that Hsp70 proteins can bind to their clients not only via short unfolded segments, but via alternative binding modes¹¹. We feel that this newly discovered, and to date only poorly characterized, binding mode will become an important research focus of the chaperone/Hsp70 field in the near future. One aspect will be to identify the substrates, which require this novel type of Hsp70 function. The interaction of Hsp70s with specific, possibly intrinsically disordered, or partially structured domains of kinases like SNF1 or Sch9 may involve canonical, as well as such novel client protein Hsp70 interactions.

2. The authors show that Ssb interacts with Sch9 in an ATP-dependent manner, suggesting

that interaction requires its chaperone function, albeit not co-translationally. This seems counter to their suggestion that it serves as a scaffold, which implies a simple structural role.

We agree with the view of the referee. The term scaffolding was misleading and we have now removed it from the text. Based on our data, we now suggest that Ssb performs like a *bona fide* chaperone, however, may employ a novel type to client protein interaction (see above).

3. If Ssb is needed for TORC1 phosphorylation of Sch9, can the *ssb* Δ effect be bypassed by a Sch9 phosphomimic? Can a phosphomimic of Maf1 bypass the defect that *ssb* Δ causes with Sch9?

This was a very good suggestion and we have performed the experiment. The results are now shown in the new Figs. 4c-e and we feel, strongly support the main conclusion of this work. The hyperactive Sch9-2D3E mutant indeed rescues Maf1 phosphorylation in cells lacking Ssb (Fig. 4d). Moreover Sch9-2D3E expressed in cells lacking Ssb partially rescues "ribosome aggregation" (Fig. 4c). However, Sch9-2D3E does not rescue the growth defects of a strain lacking Ssb (Fig. 4e). This is because Ssb has many additional functions and Sch9-2D3E rescues only with respect to loss of Sch9 activity. We have not tested if a phosphomimetic of Maf1 can bypass the Δ *ssb1* Δ *ssb2* ribosomal biogenesis defects, because besides Maf1 Sch9 has many additional targets, a number of which are also involved in ribosome biogenesis^{12,13}. We thus did not expect a clear-cut result with this approach.

4. The authors are missing opportunities to test their models by genetic interactions. For example, they argue that aggregation of ribosomal proteins in Δ *ssb* cells is due to loss of activity of Sch9. If this were true, *ssb* and *sch9* should be epistatic. Similarly, a phosphomimic of Sch9 should bypass the ribosome protein aggregation defect seen in *ssb* Δ cells. Such experimental approaches should be included.

We have now performed additional experiments with respect to "ribosome aggregation" (Fig. 4c and see also our reply to 3.). We find that with respect to the aggregation of small ribosomal subunit components loss of Ssb or Sch9 is indeed epistatic, as expected if the effect of Ssb was via Sch9. However, with respect to aggregation of large ribosomal proteins, the absence of Ssb has a more dramatic effect compared with the absence of Sch9 (Fig. 4a). The most easy explanation for these data is that Ssb acts on large ribosomal subunit biogenesis via Sch9 and in addition via a Sch9-independent pathway and likely involves the folding function of Ssb. Consistently, the hyperactive Sch9-2D3E mutant quite efficiently rescues aggregation of small ribosomal subunit proteins, however, less efficiently rescues aggregation of large ribosomal subunit proteins (Fig. 4c). These new findings are now included in the Results and Discussion.

5. L123. The authors should explain how they made the connection between Ssb and TORC1 phosphorylation of Sch9.

The starting point was the observation that both Δ *ssb1* Δ *ssb2* cells and Δ *sch9* cells display ribosome biogenesis defects. Moreover, we found the Δ *ssb1* Δ *ssb2* strain to be rapamycin sensitive, which is indicative of reduced TORC1 activity with respect to its downstream targets. The data are now included as new Fig. 2a.

6. L363 The authors report that Δ *ssb* does not induce a misfolded protein response because Hsf1 is not phosphorylated. However, they also claim that Ssb is needed for Sch9 signaling which is needed for Hsf1 phosphorylation. So, activation of Hsf1 would not be expected and cannot be used as a proxy for a misfolded protein response. The logic here is circular.

We have tried to make that point more clearly in the text. In a nutshell: Accumulation of misfolded proteins, which occurs upon heat shock leads to the phosphorylation and

activation of Hsf1. There is no difference in Hsf1 hyperphosphorylation and activation upon heat shock in the absence of Ssb/Sch9 when compared to the wild type (Fig. 6b). Thus, our data are fully consistent with the model that Sch9 activity is *not* required for the activation of Hsf1 with respect to heat shock, but is required for the activation of Hsf1 with respect to glucose starvation (Fig. 6a). We thus believe that phosphorylation of Hsf1 can be used as an indicator of misfolded protein accumulation in glucose-grown $\Delta ssb1\Delta ssb2$ and $\Delta sch9$ cells since Hsf1 phosphorylation and activation due to the accumulation of unfolded proteins is independent of Sch9 under these conditions.

List of newly included data:

Fig. 1e, left panel

SNF1-dependent phosphorylation of mutant Sch9, which lacks all TORC1-dependent sites, and the PDK1 site

Fig. 2a

rapamycin sensitivity of a $\Delta ssb1\Delta ssb2$ strain

Fig. 2d

identification of the residue (Sch9-T737), which requires Ssb for efficient TORC1-dependent phosphorylation

Fig. 3d, right panel

overexpression of Sch9 rescues Maf1 phosphorylation in the absence of Ssb

Fig. 3f-h

effect of other chaperones with respect to the Sch9-dependent phosphorylation of Maf1

Fig. 4a-b

extended analysis of ribosome aggregation in the absence of Ssb and Sch9

Fig. 4c-d

Rescue of ribosome aggregation/Maf1 phosphorylation in the absence of Ssb via expression of the permanently active Sch9-2D3E mutant

Fig. S1b

Time course of Sch9 phosphorylation upon nitrogen starvation in the presence or absence of Ssb

Fig. S2a

growth defects of single mutants in the TORC1-dependent sites of Sch9

Fig. S3e

pull-down of Ssa with FLAG-Sch9

Fig. S4b, right panel

Dephosphorylation control of hyper-phosphorylated Hsf1

Fig. S4d

Rescue of Hsf1 phosphorylation in the absence of Ssb via expression of the permanently active Sch9-2D3E mutant

Fig. S6

Validation of antibodies

References

1. Gumiero, A. *et al.* Interaction of the cotranslational Hsp70 Ssb with ribosomal proteins and rRNA depends on its lid domain. *Nat Commun*, 13563 (2016).
2. Hanebuth, M. A. *et al.* Multivalent contacts of the Hsp70 Ssb contribute to its architecture on ribosomes and nascent chain interaction. *Nat Commun* **7**, 13695 (2016).
3. Hübscher, V. *et al.* The Hsp70 homolog Ssb and the 14-3-3 protein Bmh1 jointly regulate transcription of glucose repressed genes in *Saccharomyces cerevisiae*. *Nucleic Acids Res* **44**, 5629-5645 (2016).
4. Hübscher, V., Mudholkar, K. & Rospert, S. The yeast Hsp70 homolog Ssb: a chaperone for general de novo protein folding and a nanny for specific intrinsically disordered protein domains. *Curr Genet* **63**, 9-13 (2016).
5. Urban, J. *et al.* Sch9 is a major target of TORC1 in *Saccharomyces cerevisiae*. *Mol Cell* **26**, 663-674 (2007).
6. Bonner, J. J. *et al.* Complex regulation of the yeast heat shock transcription factor. *Mol. Biol. Cell* **11**, 1739-1751 (2000).
7. Koplín, A. *et al.* A dual function for chaperones SSB-RAC and the NAC nascent polypeptide-associated complex on ribosomes. *J. Cell. Biol.* **189**, 57-68 (2010).
8. Albanese, V., Reissmann, S. & Frydman, J. A ribosome-anchored chaperone network that facilitates eukaryotic ribosome biogenesis. *J. Cell Biol.* **189**, 69-81 (2010).
9. Schmelzle, T., Beck, T., Martin, D. E. & Hall, M. N. Activation of the RAS/cyclic AMP pathway suppresses a TOR deficiency in yeast. *Mol. Cell. Biol.* **24**, 338-351 (2004).
10. Soulard, A. *et al.* The rapamycin-sensitive phosphoproteome reveals that TOR controls protein kinase A toward some but not all substrates. *Mol Biol Cell* **21**, 3475-3486 (2010).
11. Mashaghi, A. *et al.* Alternative modes of client binding enable functional plasticity of Hsp70. *Nature* **539**, 448-451 (2016).
12. Huber, A. *et al.* Sch9 regulates ribosome biogenesis via Stb3, Dot6 and Tod6 and the histone deacetylase complex RPD3L. *EMBO J* **30**, 3052-3064 (2011).
13. Huber, A. *et al.* Characterization of the rapamycin-sensitive phosphoproteome reveals that Sch9 is a central coordinator of protein synthesis. *Genes Dev.* **23**, 1929-1943 (2009).

Reviewers' Comments:

Reviewer #1 (Remarks to the Author):

The authors have satisfactorily addressed my concerns and have added significant new experimentation to answer major outstanding questions.

Reviewer #2 (Remarks to the Author):

The authors have now satisfactorily responded to my comments and have added more evidence to strengthen their conclusion.

Reviewer #3 (Remarks to the Author):

In this revision, the authors have largely addressed my concerns. One issue remains - the aggregation of ribosomal proteins is clear but whether or not they represent ribosomal or pre-ribosomal particles is not. The co-aggregation of ribosome biogenesis factors and/or ribosomal RNAs would greatly strengthen this assertion. In the absence of such data, I think the authors should be more cautious in their interpretation and refer to these as aggregated ribosomal proteins that may come from nascent proteins or from pre-ribosomal particles.

Reply to reviewers' comments

REVIEWERS' COMMENTS:

Reviewer #1 (Remarks to the Author):

The authors have satisfactorily addressed my concerns and have added significant new experimentation to answer major outstanding questions.

Reviewer #2 (Remarks to the Author):

The authors have now satisfactorily responded to my comments and have added more evidence to strengthen their conclusion.

Reviewer #3 (Remarks to the Author):

In this revision, the authors have largely addressed my concerns. One issue remains - the aggregation of ribosomal proteins is clear but whether or not they represent ribosomal or pre-ribosomal particles is not. The co-aggregation of ribosome biogenesis factors and/or ribosomal RNAs would greatly strengthen this assertion. In the absence of such data, I think the authors should be more cautious in their interpretation and refer to these as aggregated ribosomal proteins that may come from nascent proteins or from pre-ribosomal particles.

REPLY TO REVIEWERS' COMMENTS:

We again thank all 3 Reviewers for their detailed comments and their help with the manuscript. We are happy to learn that Reviewer #1 and Reviewer #2 are fully satisfied with our revision.

Reviewer #3:

We fully agree with the Referee. Co-aggregation of ribosome-biogenesis factors with ribosomal proteins is an important issue. It is thus important that Koplín et al. 2010 showed that not only ribosomal proteins, but also specifically ribosomal biogenesis factors are prone to aggregation in strains lacking Ssb/RAC. Koplín et al. state that explicitly: "*Because the detected spectrum of aggregation-prone proteins in Sse1-deficient cells is unique compared with SSB-deficient cells, we conclude that the aggregation of ribosomal proteins and ribosomal biogenesis factors is highly specific for the loss of the ribosome-associated SSB system*". These observations are fully consistent with our conclusions.

We now included this important information into the Introduction of the manuscript:

Moreover, in the absence of a functional Ssb/RAC system specifically ribosomal proteins and ribosome biogenesis factors display an enhanced tendency to aggregate^{12,13}, ...

Reviewer #3 is also right with his/her comment that the co-translational folding function of Ssb likely also contributes to the aggregation of ribosomal proteins in strains lacking Ssb/RAC. We have now made that also more clear in the Discussion:

However, our data indicate that, Ssb affects ribosome biogenesis via additional Sch9-independent mechanisms, which seemingly predominantly affect the large ribosomal subunit and may involve the folding function of Ssb.